# Rapid summer Russian Arctic sea-ice loss enhances the risk of recent Eastern Siberian wildfires

Binhe Luo [1], Dehai Luo [2,3] ✉, Aiguo Dai [4], Cunde Xiao [1] ✉, Ian Simmonds [5], Edward Hanna [6], James Overland [7], Jiaqi Shi[2,3], Xiaodan Chen [8], Yao Yao [2,3], Wansuo Duan[9], Yimin Liu [9], Qiang Zhang [10], Xiyan Xu [2,3], Yina Diao[11], Zhina Jiang[12] & Tingting Gong [13]

In recent decades boreal wildfires have occurred frequently over eastern Siberia, leading to increased emissions of carbon dioxide and pollutants. However, it is unclear what factors have contributed to recent increases in these wildfires. Here, using the data we show that background eastern Siberian Arctic warming (BAW) related to summer Russian Arctic sea-ice decline accounts for ~79% of the increase in summer vapor pressure deficit (VPD) that controls wildfires over eastern Siberia over 2004-2021 with the remaining ~21% related to internal atmospheric variability associated with changes in Siberian blocking events. We further demonstrate that Siberian blocking events are occurring at higher latitudes, are more persistent and have larger zonal scales and slower decay due to smaller meridional potential vorticity gradients caused by stronger BAW under lower sea-ice. These changes lead to more persistent, widespread and intense high-latitude warming and VPD, thus contributing to recent increases in eastern Siberian high-latitude wildfires.

Wildfires are an integral part of the Earth system and have significant impacts on human health, regional economies, biogeochemical cycles, and ecosystems[1–4]. Recent climate change has influenced both global and local wildfire patterns[5–9], and wildfires can, in turn, affect Earth's climate by releasing greenhouse gases and aerosols and altering regional soil and vegetation[10–13]. Thus, there is a complex interaction between climate and wildfires[14]. Under recent global warming, severe and widespread wildfire events have frequently occurred over the

Northern Hemisphere in spring, summer, and autumn[6,15], and emitted vast amounts of carbon dioxide ($CO_2$) and aerosols into the atmosphere, adding to air pollution. In particular, summer wildfires of boreal forests are a notable component of the global fire-induced $CO_2$ emissions[16,17].

The level of boreal wildfire activity depends on a range of surface conditions, such as heatwaves, drought, precipitation deficit, ground moisture, and lightning[18–20], which are in turn influenced by the

[1]State Key Laboratory of Earth Surface Processes and Resource Ecology, Beijing Normal University, Beijing 100875, China. [2]Key Laboratory of Regional Climate-Environment for Temperate East Asia, Institute of Atmospheric Physics, Chinese Academy of Science, Beijing 100029, China. [3]University of Chinese Academy of Sciences, Beijing 101408, China. [4]Department of Atmospheric and Environmental Sciences, State University of New York, Albany, NY, USA. [5]School of Geography, Earth and Atmospheric Sciences, University of Melbourne, Melbourne, VIC, Australia. [6]Department of Geography, School of Life and Environmental Sciences, University of Lincoln, Lincoln, UK. [7]NOAA/Pacific Marine Environmental Laboratory, Seattle, WA, USA. [8]Department of atmospheric and oceanic sciences, Fudan University, Shanghai 200438, China. [9]State Key Laboratory of Numerical Modeling for Atmospheric Sciences and Geophysical Fluid Dynamics (LASG), Institute of Atmospheric Physics, Chinese Academy of Sciences, Beijing 100029, China. [10]Department of Earth system science, Tsinghua University, Beijing 100084, China. [11]College of Oceanic and Atmospheric Sciences, Ocean University of China, Qingdao 266101, China. [12]Institute of Global Change and Polar Meteorology, Chinese Academy of Meteorological Sciences, Beijing 100081, China. [13]Key Laboratory of Ocean Circulation and Waves, Institute of Oceanology, Chinese Academy of Sciences, Qingdao 266400, China. ✉e-mail: ldh@mail.iap.ac.cn; cdxiao@bnu.edu.cn

overlying atmospheric circulation patterns[21–23]. The increase in air temperatures and decrease in precipitation or relative humidity have been shown to be two key drivers of boreal wildfires[18]. Besides North America, Siberia is another hotspot for summer boreal wildfires and $CO_2$ emissions in recent decades[24–27]. The burned areas and fire-induced $CO_2$ emissions show significant increases over boreal Eurasia (mainly over Siberia) since 2000[16], while boreal wildfires over North America show large interannual variability but no significant trends[17]. Even within Siberia, the increases in wildfire frequency and areal extent are non-uniform due to regional decreases in precipitation and soil moisture[28,29]. From 2003–2020 summer wildfires show a significant increasing (decreasing) trend over eastern (western) Siberia[22]. In 2021, boreal wildfires over eastern Siberia generated the largest $CO_2$ emissions of 2000-2021, amounting to more than 150% of the annual mean emissions[17]. Forest fires and $CO_2$ emission in Russia are also linked to internal atmospheric variability such as Arctic Oscillation[22] and atmospheric blocking events[30,31]. Furthermore, reduced ground-moisture content associated with enhanced summer warming in the Russian Arctic may favor wildfires in eastern Siberia[28].

While the summer Arctic warming on interannual timescales is partly linked to atmospheric forcing[32], its long-term (≥decadal timescales) trend is mainly related to the declining of summer Arctic sea-ice[33] due to increasing $CO_2$[34] and decadal Arctic upper ocean warming[35,36]. In addition to the effect of increasing $CO_2$, decadal variability of summer Arctic sea-ice has been found to be linked to the Atlantic Meridional Overturning Circulation (AMOC) associated with the Atlantic Multidecadal Oscillation (AMO) and Pacific Decadal Oscillation (PDO)[37]. Because the summer Arctic sea-ice decline mainly occurs over the eastern Siberian side of Russia[38], one could surmise that the summer Arctic warming related to Russian Arctic sea-ice decline could significantly contribute to increased wildfires over eastern Siberia via increased high-latitude Siberian lightning[20], decreased precipitation and soil moisture[24,29]. However, it is unclear whether summer Arctic warming associated with Russian Arctic sea-ice decline or internal atmospheric variability is more influential in driving the increases in eastern Siberian wildfires in recent decades. To our knowledge, no such study has been conducted in past years. As a key meteorological variable, the vapor pressure deficit (VPD) is a very important factor that influences fuel aridity and fire behavior[39] ("Methods" section). The VPD includes the effect of air temperature and relative humidity (or saturation vapor pressure) and is associated with atmospheric circulation patterns[22,40], and it explains more of the variance in fire activity than does precipitation, drought indices, air temperature, and wind individually. It is also more successful in explaining the ignition, spread, intensity, and size of forest fires than these other meteorological variables[39,40]. The high air temperature and low relative humidity or precipitation deficit give rise to high VPD that increases extreme fire risk[39]. Winds from hot inland areas and subsidence related to high-pressure systems also generate hot and dry air, leading to high VPD values[40]. Thus, VPD is of relevance in studies of the links of wildland fires to meteorological conditions[39,40].

In this study, we analyze satellite and reanalysis data using an approach ("Methods" section) to quantify different contributions from the background Arctic warming and internal atmospheric variability related to changes in Siberian blocking events to the recent wildfire risk trend in eastern Siberia by calculating VPD. In this approach, the summer mean of daily surface (2 m) air temperature (SAT) anomalies without Siberian blocking events (which have lifetimes of 10–20 days) can be regarded as the summer background eastern Siberian Arctic warming (BAW) related to Russian Arctic sea-ice loss. By comparing the slope rates of the summer eastern Siberian SAT and VPD over 2004-2021 with and without blocking events ("Methods" section), we further estimate the relative contributions from the BAW and Siberian blocking events which result primarily from atmospheric internal variability although it is possible that these events may also have been influenced

by the recent BAW. We also investigate how a wildfire event depends on the evolution of Siberian blocking under different SIC conditions via changes in the persistence, zonal scale, movement, decay, and latitudinal location of Siberian blocking events and VPD based on a daily composite.

## Results

### Variation and trend of eastern Siberian wildfires and their linkages to summer Russian Arctic sea-ice loss

The Fire Weather Index (FWI) is a numerical rating of fire intensity, which is based on the initial spread index, and the buildup index that is a numerical rating of the total amount of fuel available for combustion[18,29]. The index data, which is available from the Copernicus Emergency Management Service for the European Forest Fire Information System (EFFIS), has been widely used to evaluate fire danger due to weather and climate variations across the globe[29,41–43]. Here, we use the FWI data.

Our results show that during 1979–2021 the summer (June to August, JJA) mean time series of the daily FWI anomaly averaged over eastern Siberia (90°–150°E; 60°–75°N) (Fig. 1a) has a significant negative correlation of −0.36 ($p < 0.05$) with the JJA-mean sea-ice concentration (SIC) anomaly averaged over the Russian Arctic region (30°–130°E; 65°–85°N) during 1979–2021 (Fig. 1b) based on non-detrended data (Fig. 1a, b). Their correlation coefficient has a greater magnitude of −0.56 ($p < 0.01$) over 2004–2021, but a smaller magnitude of −0.03 over 1979–2003. This indicates that the linkage of summer eastern Siberian wildfires to the Russian Arctic SIC variability is notably intensified over 2004–2021, a result which also holds for the Global Fire Assimilation System (GFAS) data (Supplementary Fig. 1a) because there is a similar negative correlation of −0.55 ($p < 0.01$) between the wildfire faction and SIC. On the other hand, the summer FWI over 2004–2021 shows an upward trend consistent with the trends of wildfire fraction, burned area, and burned fraction over eastern Siberia for the GFAS, MODIS and Global Fire Emissions Database (GFED) data (Supplementary Fig. 1), indicating that increased wildfires with a poleward expansion indeed occur over eastern Siberia in recent two decades. Moreover, all the data results show a substantial shift toward a mega wildfire regime over eastern Siberia from 2011 to 2021, while the transition of eastern Siberian wildfires trend takes place at about 2004 (Fig. 1a) concurrent with the SIC trend transition occurring at 2004 (Fig. 1b).

We note that when the FWI and SIC data over 1979–2021 are detrended, two of these correlations weaken to −0.27 ($p < 0.05$) for 1979–2021 and −0.42 ($p < 0.05$) for 2004–2021, while that for 1979–2003 increases in magnitude to −0.20. This implies that the enhanced Arctic amplification associated with $CO_2$-induced warming and interdecadal oceanic variability can strengthen the connection of recent summer wildfires over eastern Siberia to the Russian Arctic SIC decline. In fact, the increase in recent eastern Siberian wildfires (Fig. 1e) is also linked to an enhanced anticyclonic circulation anomaly trend over eastern Siberia (Fig. 2e). Thus, our emphasis in this study is to evaluate the relative contributions of the BAW associated with the summer Russian Arctic SIC decline and internal atmospheric variability to the summer wildfire trend over eastern Siberia over 2004–2021.

When the (non-detrended) data are normalized, we see that the summer FWI over eastern Siberia shows a significant increasing trend with the slope rate of 1.18 standard deviations (STDs)/decade ($p < 0.01$) over 2004–2021 (Fig. 1a, dashed red curve), whereas the declining of the Russian Arctic SIC over 2004–2021 (Fig. 1b, dashed red curve) has a slope rate of −1.04 STDs/decade ($p < 0.01$). Thus, one could infer that the recent increasing trend of the FWI over eastern Siberia is likely associated with the declining of the recent Russian Arctic SIC possibly via the increase in Arctic warming over eastern Siberia. However, the recent decline of the summer Russian Arctic SIC is not driven by a single factor[44]. Instead, it is mainly driven by a combination of

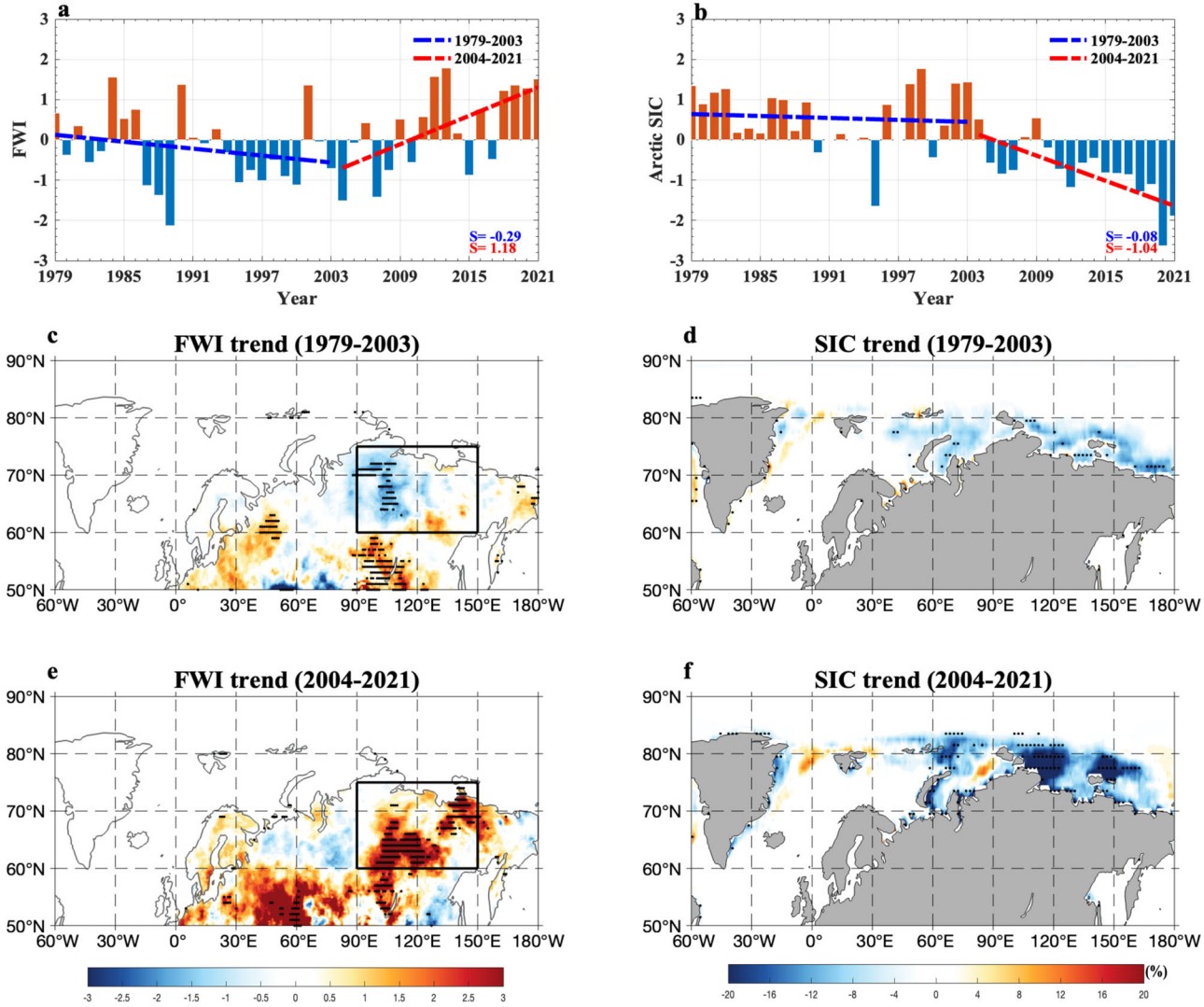

**Fig. 1 | Temporal variations of the summer fire weather index (FWI) over eastern Siberia and sea-ice concentration (SIC) anomaly over the Russian Arctic during 1979–2021 and their linear trend patterns over 1979–2003 and 2004–2021. a, b** Normalized time series of summer (June–August, JJA) mean (**a**) fire weather index (FWI, unit: non-dimensional) anomaly averaged over eastern Siberia (90°–150°E, 60°–75°N) and (**b**) JJA-mean sea-ice concentration (SIC, unit: %) anomaly averaged over the Russian Arctic region (30°–130°E; 65°–85°N) during 1979–2021, where the dashed blue (red) line represents the linear trends of the FWI and SIC over 1979–2003 (2004–2021) with the slope rates of −0.29 and −0.08 (1.18 and −1.04) standard deviations (STDs) per decade. **c**–**f** Linear trend patterns of JJA-mean **c, e** FWI (color shading, unit: non-dimensional value per decade) and **d, f** SIC (color shading, unit: % per decade) anomalies over **c, d** 1979–2003 and **e, f** 2004–2021. The black box denotes the eastern Siberia (90°–150°E, 60°–75°N). The dotted regions represent that the linear trends are statistically significant ($p < 0.05$) for the Mann–Kendall test.

global warming or $CO_2$-induced warming and internal multidecadal variability[45–47] primarily via oceanic processes[35], which is non-uniform and mainly located in the Russian Arctic from the Barents Sea to East Siberia[46] due to enhanced Arctic upper ocean heat content since the 2000s[35,36] via intensified ocean heat transports (OHTs) into the Russian Arctic[32,45].

While the main purpose of our study is not to explore the cause of the summer Arctic SIC loss, our result shows that the JJA-mean upper OHT averaged from the sea surface to the depth 150 m ("Methods" section) and over the location 30°E, 65°–85°N near the Barents Sea Opening (BSO) (Supplementary Fig. 2) exhibits an increasing trend over 1979–2018 and has a simultaneous significant negative correlation of −0.51 ($p < 0.01$) with the Russian Arctic SIC. Thus, the recent loss of the summer Russian Arctic SIC is related to enhanced BSO OHT via upper ocean warming[45] in the Arctic which mainly stems from the Atlantic warm water[47]. The intensified BSO OHT is due to the positive AMO in large part and negative PDO in small part (Supplementary Fig. 3), as demonstrated by climate modeling experiments[37]. Thus, the positive phase of AMO and the negative phase of PDO would appear to combine to contribute to the summer Russian Arctic SIC decline over 2004–2021 by strengthening the upper OHT to the Russian Arctic[37]. Consequently, increasing $CO_2$, and the AMO and PDO jointly dominate the Arctic warming and the declining trend of the Russian Arctic SIC in recent decades[47–49], although the Arctic warming is much weaker in summer than in winter[33,34].

While climate models can capture the climatological Arctic warming caused by sea-ice loss[50], they are much less successful in capturing the blocking frequency especially in high latitudes[51]. Thus, we would not be confident to use the climate models to estimate the contribution of internal atmospheric variability associated with changes in Siberian blocking events to summer Siberian wildfires. Here, we present an approach to evaluate the relative contributions of the BAW trend associated with the Russian Arctic SIC loss and internal atmospheric variability related to changes in Siberian blocking events

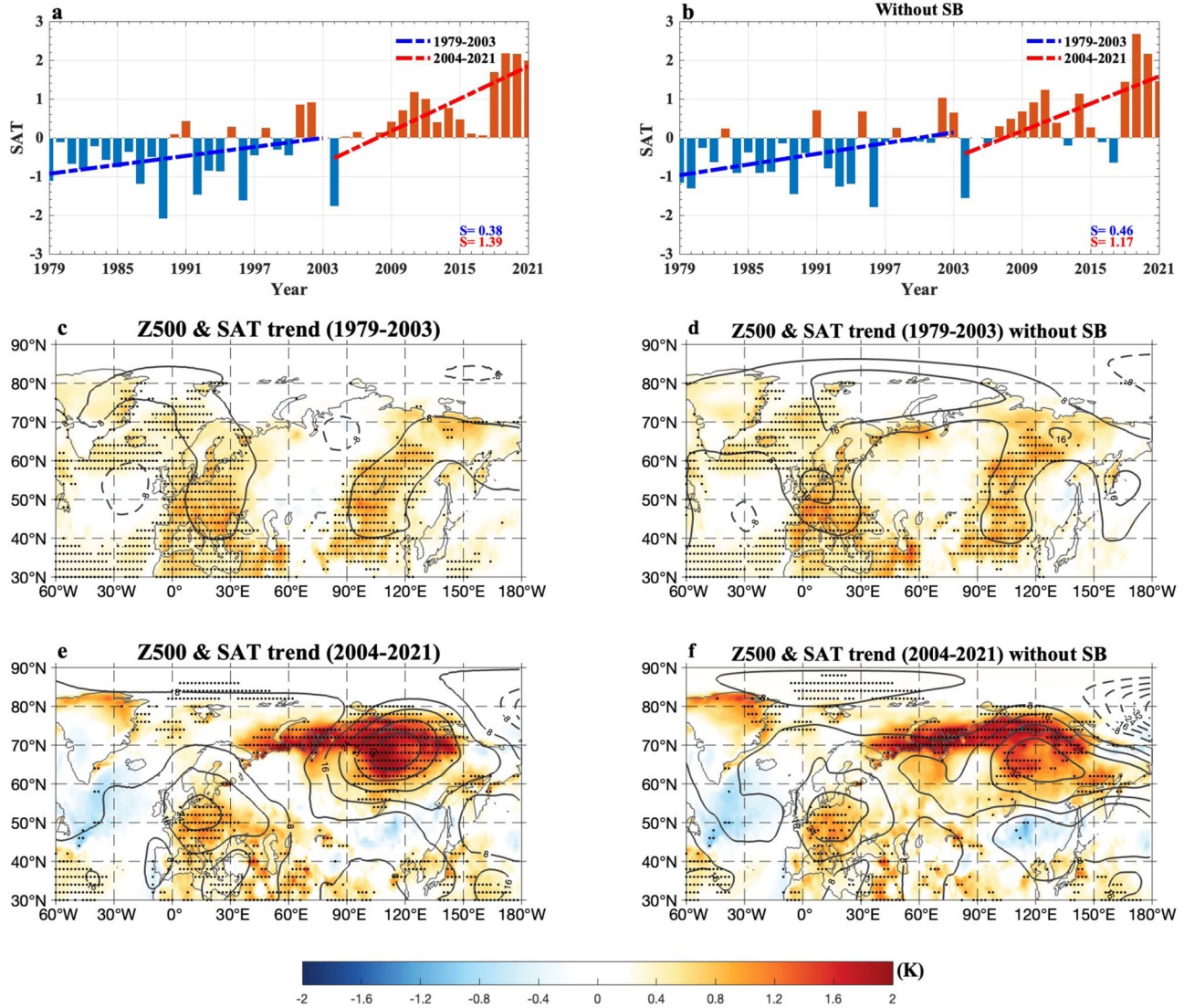

**Fig. 2 | Temporal variations of the summer surface air temperature (SAT) anomaly over eastern Siberia during 1979–2021 and linear trend patterns of the summer SAT and 500-hPa geopotential height (Z500) anomalies over 1979–2003 and 2004–2021 with and without Siberian blocking (SB) events.** **a**, **b** Normalized time series of summer (June–August, JJA) mean surface air temperature (SAT, unit: K) averaged over eastern Siberia (90°–150°E, 60°–75°N) during 1979–2021 based on the daily-mean SAT of the ERA5 data, where the dashed blue (red) line represents the linear trends over 1979–2003 (2004–2021) with the slope rates of 0.38 and 0.46 (1.39 and 1.17) standard deviations (STDs) per decade for the cases **a** with and **b** without Siberian blocking (SB) events (the case without SB events represents that blocking days from lag −10 to 10 days are removed for each SB event and lag 0 denotes the peak day of SB). **c**–**f** Linear trend patterns of JJA-mean SAT (color shading, unit: K per decade) and 500-hPa geopotential height (Z500, contour interval= 8, unit: gpm per decade) anomalies over **c**, **d** 1979-2003 and **e**, **f** 2004–2021 for the cases **c**, **e** with and **d**, **f** without SB events, where the dotted regions represent that the trends are statistically significant (*p* < 0.05) for the Mann–Kendall test.

to increases (or trends) in summer eastern Siberian wildfires over 2004–2021 by comparing the slope rates of the JJA-mean eastern Siberian SAT and VPD time series over 2004–2021 with and without Siberian blocking events ("Methods" section). Here, we excluded the blocking days from lag −10 to 10 days of each blocking event (lag 0 denotes the peak day of blocking) in summer over Siberia (90°--120°E) identified by the blocking index ("Methods" section) to calculate the JJA-mean values of remaining daily SAT and VPD as the JJA-mean SAT and VPD without the effect of Siberian blocking events.

During 2004–2021 strong positive wildfire trends appeared in a widespread region extending from eastern Europe south of 60°N to the high latitudes of eastern Siberia north of 60°N (Fig. 1e), which we have argued is a consequence of amplified Arctic warming and, in turn, is related to strong Russian Arctic SIC decline over 2004–2021 (Fig. 1f). In contrast, only weak wildfire trends appeared over Europe and central Eurasia south of 60°N during 1979–2003 (Fig. 1c), during a time of

the positive SIC anomaly over the Russian Arctic (Fig. 1b) with its weak declining trend (Fig. 1d). The wildfire trend over eastern Siberia is also influenced by enhanced drought and decreased precipitation under recent global warming[52]. Thus, the poleward migration and increasing trend of the eastern Siberian wildfires over 2004–2021 may be linked to the rapid declining of summer Russian Arctic SIC.

Our result based on the normalized JJA-mean time series of the daily-mean SAT anomaly averaged over eastern Siberia (90°–150°E; 60°–75°N) derived from the European Centre for Medium-Range Weather Forecasts Reanalysis 5 (ERA5) global reanalysis data[53] shows that the Russian Arctic warming over eastern Siberia has a linear increasing trend with a slope rate of 1.39 STDs/decade (*p* < 0.01) over 2004–2021 (Fig. 2a). Thus, it is possible that the increasing trend of recent eastern Siberian wildfires during 2004–2021 is due to both enhanced eastern Siberian Arctic warming and decreased precipitations over eastern Siberia (Fig. 3a, c). We also note that the eastern

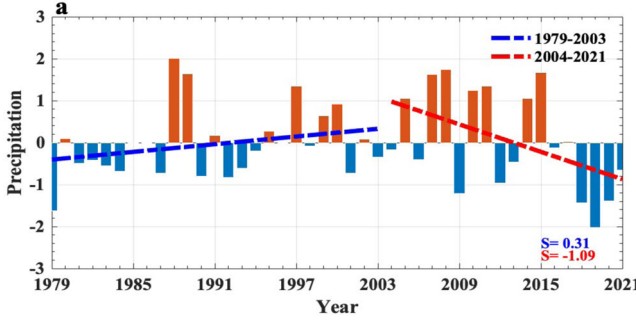

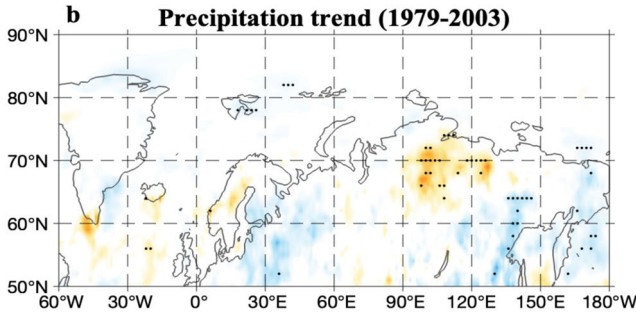

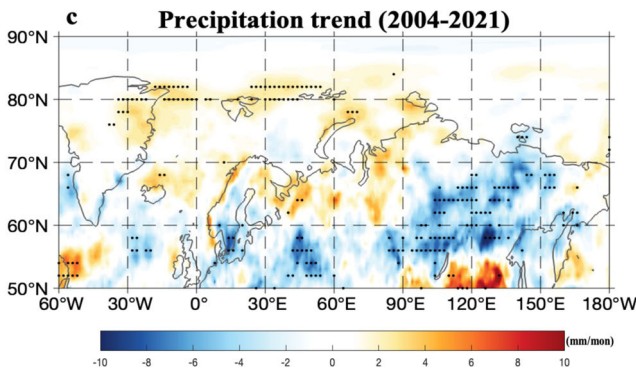

**Fig. 3 | Time series of summer precipitation anomaly over eastern Siberia during 1979–2021 and its linear trend patterns over 1979–2003 and 2004–2021. a** Normalized time series of summer (June to August, JJA) mean precipitation anomaly (unit: mm/month or mm/mon) averaged over eastern Siberia (90°–150°E, 60°–75°N) during 1979–2021 with the slope rates of the linear trends (unit: non-dimensional value per decade) over 1979–2003 (dashed blue line) and 2004–2021 (dashed red line). **b, c** Linear trend patterns of summer mean precipitation (unit: mm/mon per decade) anomalies over (**b**) 1979–2003 and **c** 2004–2021, where the dot represents the significant region with a 95% confidence level for a two-sided student t-test.

Siberian Arctic SAT time series and Russian Arctic SIC show an opposite variation over the total period of 1979–2021 (Supplementary Fig. 4), with a strong (weak) correlation of −0.74 (−0.12) over 2004–2021 (1979–2003). These correlations are −0.57(−0.30, $p < 0.05$) when detrended data are used. This reflects a strengthened connection of the SAT increase in the eastern Siberian high latitudes to the Russian Arctic SIC decline during 2004–2021 associated with upper ocean warming over the Russian Arctic[35] via enhanced radiative heating due to summer sea-ice melting[54]. If the daily maximum SAT ($SAT_{max}$ or $T_{max}$) is used, the JJA-mean $SAT_{max}$ over eastern Siberia (Supplementary Fig. 5) has a high correlation of 0.99 ($p < 0.01$) with the JJA-mean SAT obtained from the daily-mean SAT (Fig. 2a). Thus, one can well use the daily-mean (or daily) SAT to depict Arctic warming over eastern Siberia. Our results further reveal that the summer FWI and SAT over eastern Siberia have a positive correlation coefficient of 0.72 ($p < 0.01$) (0.33, $p < 0.05$) over 2004–2021 (1979–2003)

for non-detrended data. Clearly, their correlation is significantly intensified from 1979–2003 to 2004–2021. Such a correlation is not altered by using daily maximum SAT. Thus, an intensified connection of eastern Siberian wildfires to SATs over eastern Siberian Arctic during 2004–2021 is probably due to an intensified SIC loss leading to enhanced BAW. In contrast, this connection is less strong in the 1979–2003 period because the SIC decline is weak. Consequently, it is inferred that the rapid SIC loss can significantly strengthen the influence of eastern Siberian Arctic SAT anomalies on recent summer eastern Siberian wildfires.

Our simulations from a fully coupled climate model (namely, the CESM1, see "Methods" section) with and without fixed SIC[34] indicate that the increase in the summer Arctic warming is mainly caused by the Arctic sea-ice loss (Supplementary Fig. 6), which is consistent with the ensemble result from the Polar Amplification Model Intercomparison Project contribution (PAMIP) to the sixth Coupled Model Intercomparison Project (CMIP6)[50] (Supplementary Fig. 7). This simulation result is also supported by a 100-member ensemble atmospheric general circulation model (AGCM) experiment with increased concentrations of greenhouse gases, ozone, and aerosols[55] and the experiments of six atmosphere–ocean coupled general circulation models (GCMs)[56]. Thus, the BAW is mainly produced by the summer Russian Arctic SIC loss associated with the external forcing[55,56] and Arctic upper ocean warming[35].

We find that the JJA-mean eastern Siberian Arctic warming trend has a smaller slope rate of 1.17 STDs/decade ($p < 0.01$) over 2004–2021 when these Siberian blocking (SB) events are removed (Fig. 2b), which represents the BAW trend. Clearly, the presence of summer SB events as an internal mode of atmospheric variability can induce an increase of 0.22 STDs/decade in the upward trend of the summer Arctic warming in the high latitudes of eastern Siberia from 1.17 STDs/decade (Figs. 2b) to 1.39 STDs/decade (Fig. 2a). The high latitude Arctic warming and anticyclonic anomaly trends are still strong over eastern Siberia even for the case without SB events (Fig. 2f), which are mainly produced by the declining of the Russian Arctic SIC (Fig. 1f), as corroborated by the model experiments (Supplementary Figs. 6 and 7) and previous model results[55,56]. We also see that without blocking events the linear trend of eastern Siberian Arctic warming over 2004–2021 becomes closer to the opposite value of the SIC trend (−1.04 STDs/decade). Thus, this JJA-mean high-latitude eastern Siberian Arctic warming trend without the effect of SB events (Fig. 2f) may be considered as the BAW trend mainly due to the Russian Arctic SIC loss. Therefore, the trend of eastern Siberian Arctic warming over 2004–2021 (Fig. 2e) consists of the BAW trend (Fig. 2f) and a trend induced by the internal atmospheric variability, while SB events can be modulated by the slow change in the BAW based on the blocking theory ("Methods" section).

Because high VPD corresponds to a fire-prone area and controls the severity and burned area of boreal forest wildfires[57] via ignition and spread[58], the value of VPD can be used to characterize recent wildfire risk[18,40]. This is justified because the summer VPD has a significant correlation of 0.89 (0.85) ($p < 0.01$) with the summer FWI over 2004-2021 (1979–2003). Therefore, the linear slope rate of the summer eastern Siberian VPD time series over 2004−2021 can be selected as a proxy indicator to measure the recent trend of observed eastern Siberian wildfires. Our VPD result based on daily-mean SAT and dew point temperature from the ERA5 data shows that the summer VPD exhibits a significant increase over 2004−2021 (Fig. 4a). Its trend pattern (Fig. 4e) over 2004−2021 closely matches that of FWI (Fig. 1e), even though the increasing trend of VPD is less when SB events are removed (Fig. 4f). Thus, the joint of the BAW trend and SB-induced additive warming is conducive to increases in boreal forest fires over eastern Siberian high latitudes (Fig. 1e) via decreased precipitation (Fig. 3c) and increased VPD (Fig. 4e), despite a fact that BAW favors increased Arctic lightning[20,59] due to increased convective available

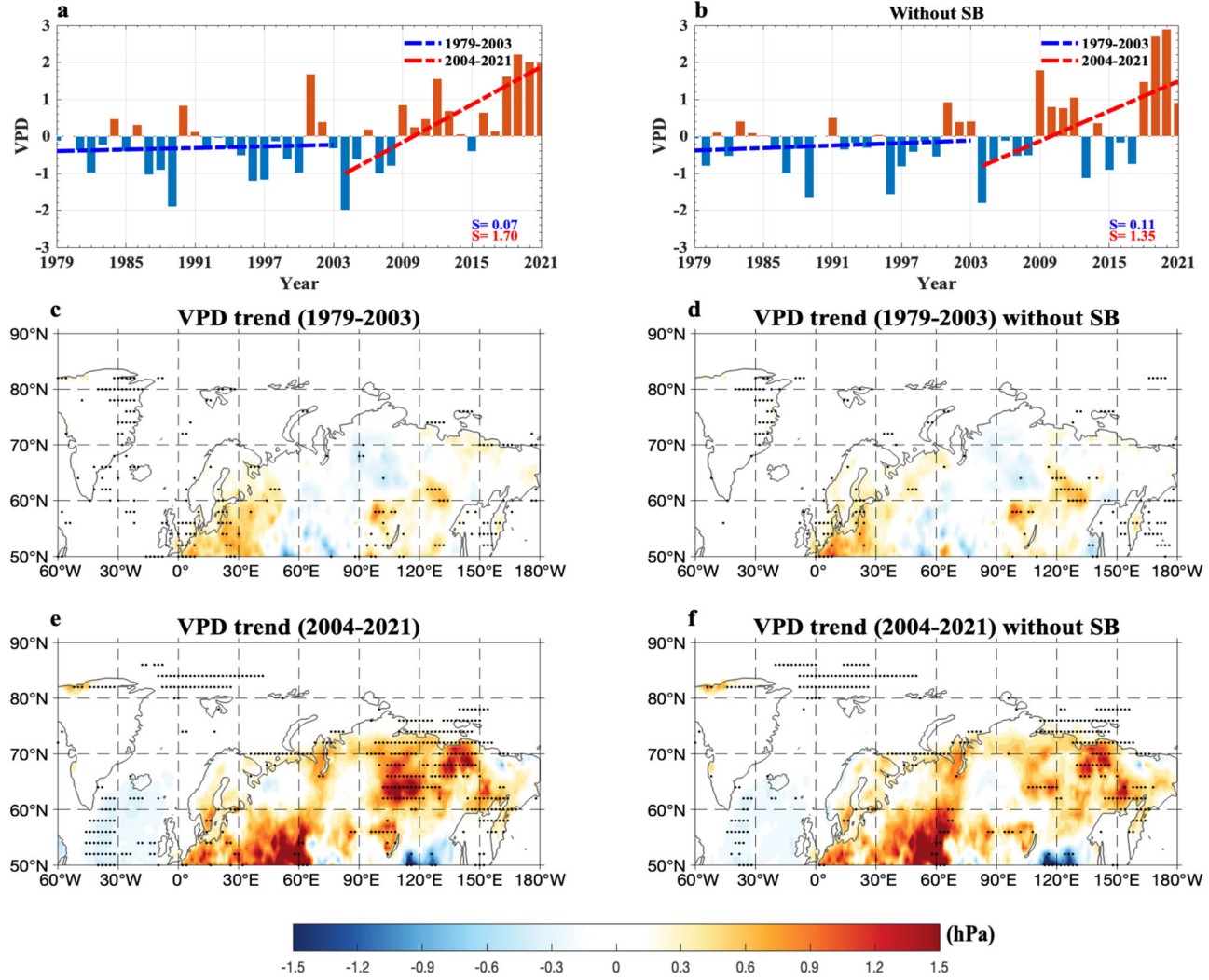

**Fig. 4 | Temporal variations of summer vapor pressure deficit (VPD) over eastern Siberia during 1979–2021 and its linear trend patterns over 1979–2003 and 2004–2021 with and without Siberian blocking (SB) events. a, b** Normalized time series of summer (June–August, JJA) mean vapor pressure deficit (VPD, unit: hPa) averaged over eastern Siberia (90°–150°E, 60°–75°N) during 1979–2021 based on the daily-mean surface air temperature (SAT) from the ERA5 data, where the dashed blue (red) line represents the linear trends of VPD over 1979–2003 (2004–2021) with the slopes of 0.07 and 0.11 (1.70 and 1.35) standard deviations (STDs) per decade for the cases **a** with and **b** without Siberian blocking (SB) events (the case without SB events represents that blocking days from lag − 10 to 10 days are removed for each SB event and lag 0 denotes the peak day of SB). **c–f** Linear trend patterns of JJA-mean VPD (color shading, unit: hPa per decade) over **c, d** 1979–2003 and **e, f** 2004–2021 **c, e** with and **d, f** without SB events based on the daily-mean SAT, where the dotted region represents that the trends are statistically significant ($p < 0.05$) for the Mann–Kendall test.

potential energy[20], soil moisture deficit and shrinking snow cover over eastern Siberia[60,61].

We find that the summer VPD has a significant correlation of 0.89 (0.68) ($p < 0.01$) with the eastern Siberian SAT over 2004–2021 (1979–2003), indicating that the variability of the JJA-mean VPD is mainly controlled by the eastern Siberian SAT. However, the summer VPD has no significant correlation ($-0.07$, $p > 0.1$) with the Arctic SIC over 1979–2003, while it exhibits a significant negative correlation of $-0.69$ ($p < 0.01$) with the SIC over 2004–2021. This indicates that the increasing trend of VPD over 2004–2021 likely stems from the rapid loss of the summer Russian Arctic SIC. In general, enhanced BWA corresponds to a decreased ground moisture extent[61] due to reduced precipitation (Fig. 3c), which can increase the likelihood of lightning strikes leading to wildfires[28]. Thus, the major trend of eastern Siberian wildfires is dominated by the BAW trend (Fig. 2f) because the increasing trend of VPD is dominant even when SB events are removed (Fig. 4f). Because there is a nonlinear dependence of VPD on the SAT[39], the JJA-mean VPD time series over eastern Siberia (Fig. 4a, b) shows

a stronger increasing trend than that of the JJA-mean SAT time series over 2004–2021 (Fig. 2a, b), which has a linear slope of 1.70 (1.35) STDs/decade with $p < 0.01$ for the case with (without) SB events. Such a VPD trend change reflects the different contributions of enhanced BAW and changes in SB events to the increase in recent eastern Siberian wildfires.

### Different contributions of summer Russian Arctic sea ice loss and Siberian blocking events to the increasing trends of eastern Siberian Arctic warming and vapor pressure deficit

Because Siberian blocking or SB can induce an increase of 0.22 STDs/decade in the summer eastern Siberian SAT trend over 2004–2021, we can estimate that about 84% (1.17/1.39≈0.84) of the warming trend over eastern Siberia directly comes from the Russian Arctic SIC decline via enhanced Arctic warming, but the remaining 16% (0.22/1.39 ≈ 0.16) likely stems from decadal changes in SB events ("Methods" section). If the daily maximum SAT (SAT$_{max}$) is used, one can estimate that the BAW (SB events) contributes to ~85% (~15%) of the eastern Siberian

warming trend over 2004–2021 (Supplementary Fig. 8). Thus, the results of warming trends obtained from daily maximum and daily-mean SATs are very close. However, as further revealed by Fig. 4a, b, SB events can cause an additional increase of 0.35 STDs/decade in the summer VPD over 2004–2021 based on daily-mean SAT data. Thus, it is estimated that ~79% of the increasing trend of the summer VPD over 2004–2021 stems from the BAW trend, whereas its remaining ~21% comes from decadal changes in SB events. Clearly, decadal changes in SB events have a stronger effect on the upward trend of VPD than on the rising trend of the eastern Siberian SAT during 2004–2021. Consequently, we infer that ~79% (~21%) of the increasing trend of eastern Siberian wildfires over 2004–2021 comes from the BWA (changes in SB events) via changes in VPD[39].

If the daily maximum SAT ($SAT_{max}$) is used instead of the daily-mean SAT, the spatial pattern of the obtained summer-mean VPD trend (Supplementary Fig. 8) matches that from the daily-mean SAT (Fig. 4) very well. Thus, there is no clear difference in the VPD pattern between $SAT_{max}$ and daily-mean SAT. It is also estimated that the BAW (SB events) contributes to ~84% (~16%) of the increasing trend of VPD over 2004–2021 when the $SAT_{max}$ is used. Clearly, the role of SB events is slightly weaker in the increase in VPD for the $SAT_{max}$ than for the daily-mean SAT. Thus, it is likely that increased nighttime VPDs and night-time fires associated with SB events play a role in the increasing trend of eastern Siberian wildfires[62] probably because the daily-mean data includes the effect of nighttime wildfires. At the same time, we can also see that the summer mean effect of SB events also strengthens the summer-mean anticyclonic anomaly and Arctic warming over eastern Siberia (Fig. 2e) compared to that without SB events (Fig. 2f). Thus, in recent decades the overall increasing trends of summer eastern Siberian warming and VPD due to the BAW trend mainly related to Russian Arctic SIC loss can be amplified by changes in SB events, which is a combined consequence of the BAW trend and slow changes in SB events. Of course, the Russian Arctic SIC decline over 2004–2021 plays a much larger role than SB events.

Our results show that over 2004–2021 the slope rate of the decrease trend of the summer precipitation over eastern Siberia (−1.09 STDs/decade, Fig. 3a) is close to the opposite value of the BAW trend (1.17 STDs/decade, Fig. 2b), and these two series have a correlation of −0.44 ($p < 0.05$). Due to the presence of an anticyclonic anomaly trend over eastern Siberia (Fig. 2f), increased surface heating associated with subsidence and increased solar radiation mainly take place over eastern Siberian high latitudes due to reduced cloudiness, which can cause a significant decrease in precipitation over eastern Siberia over 2004–2021 (Fig. 3a, c). This negative trend is weak over eastern Siberia over 1979–2003 (Fig. 3b) because the BAW trend is not strong. A significant increase in the summer VPD trend over 2004–2021 (Fig. 4f) is also seen in a widespread region of eastern Siberia possibly due to the presence of enhanced anticyclonic anomaly and BAW, even if SB events are absent. To some extent, a significant increase in the BAW resulting from the SIC loss is a major factor leading to decreased precipitation and increased VPD over eastern Siberia. In fact, the terrestrial aridity over eastern Siberia is aggravated due to enhanced land warming in response to global warming[63], even though the soil moisture deficit related to decreased precipitation or drought can also feedback to global warming via soil moisture–atmosphere coupling[64].

Intensified zonal westerly winds are also seen in the anticyclonic region north of 65°N (Supplementary Fig. 9) due to the combined effect of the enhanced stationary anticyclonic anomaly associated with the BAW and SB events, which promotes the growth of ignited fires and fire spread[65] toward the northeast side (Fig. 1e). Thus, the recent BAW trend and associated anticyclonic anomaly trend pattern provide a favorable environment for increased eastern Siberian wildfires via enhanced VPD associated with reduced relative humidity and decreased precipitation or reduced soil moisture[28]. This explains why the BAW trend dominates the increasing trend in eastern Siberian

wildfires. However, no intense warming is seen over eastern Siberia during 1979–2003 (Fig. 2d) even when the effect of SB events is included (Fig. 2c). Such a weak warming is not conducive to increases in intense wildfire events occurring over eastern Siberian high latitudes (Fig. 1c).

## Role of Siberian blocking events in the internal variability of eastern Siberian wildfires

While the slow variation of SB events plays a secondary role in the increasing trend of eastern Siberian wildfires over 2004–2021, they may reflect the role of internal atmospheric variability in eastern Siberian wildfires. Below we establish that SB events lead to wildfire events over eastern Siberia by generating extremely high temperatures and high VPDs and such an effect is strong under low SIC conditions. We further remove the linear trends of the Russian Arctic SIC, FWI, and meteorological variables (e.g., SAT, Z500, VPD, etc.) during 1979-2021 to retain the role of internal atmospheric variability associated with SB events.

We find there to be 76 SB events during 1979–2021 based on the blocking index ("Methods" section), with 47 of these falling in 1979–2003 and 29 in 2004–2021, corresponding to similar frequencies of 1.88 and 1.61 events per summer, respectively. Hence, SB events did not occur more frequently during 2004–2021 than during 1979–2003. To explore this we further define the −0.5 (0.5) STDs of the detrended summer Russian Arctic SIC anomaly during 1979–2021 as a threshold of the low (high) SIC. We see that there are 22 (27) Siberian blocking events in the 12 (12) low (high) SIC summers over 1979–2021. Thus, the low (high) SIC corresponds to 1.83 (2.25) SB events per summer, indicating that the Russian Arctic SIC decline does not favor the number of SB events. Hence, the recent increase in eastern Siberian wildfires cannot be explained in terms of changes in the number of summer SB events. Although SB reflects a sub-seasonal internal atmospheric variability, it does not imply that SB is not forced by the BAW trend related to the Russian Arctic SIC decline. While the Russian Arctic SIC decline cannot directly increase the frequency (event number) of SB events, the SB events and associated wildfires can be influenced by the Russian Arctic sea-ice loss (Fig. 1f) via the slow change in BAW (Fig. 2f).

The time-mean composite daily SAT and 500-hPa geotential height (Z500) anomalies averaged over the blocking lifecycle (from lag −10 to 10 days) of SB events ("Methods" section) show that SB events have a large (small) zonal scale and occur in the high (low) latitude eastern Siberia for the low (high) SIC summer during 1979–2021 (Fig. 5a, b). Thus, under low (high) SIC conditions SB events lead to high (low) VPD (Fig. 5c, d) that favors intense (weak) wildfire events with a large (small) area in the higher (lower) latitude sides of eastern Siberia during the blocking episode (Supplementary Fig. 10), in agreement with the composite results of the daily wildfire fraction based on the GFAS data during 2003–2021 (Supplementary Fig. 11). This can happen because decreased precipitation (Fig. 6) and increased VPD (Fig. 5) during the blocking episode are much greater in the eastern Siberian high latitudes under low SIC conditions (Figs. 5c and 6a) than under extensive SIC (Figs. 5d and 6b). In fact, intense heatwaves and drought are favored in the eastern Siberian high latitudes due to the presence of SB[66] under low SIC conditions[67]. Thus, the weather conditions associated with SB events under a low SIC condition favor the outbreak and poleward migration of strong wildfire events over eastern Siberia.

On the other hand, we note that SB events are more persistent and show a slower decay for the low SIC (Fig. 5e) than for the high SIC (Fig. 5f). We further see that the lifetime of SB confined in the region 90°–150°E is about 9 (6) days for the low (high) SIC (using 40 gpm as the blocking amplitude threshold). Because the SB has larger zonal scale and longer lifetime for the low SIC (Fig. 5a, e) than for the high SIC (Fig. 5b, f), it inevitably leads to more persistent, intense and widespread wildfire events over eastern Siberia (Supplementary Fig. 10a, c) due, in turn, to

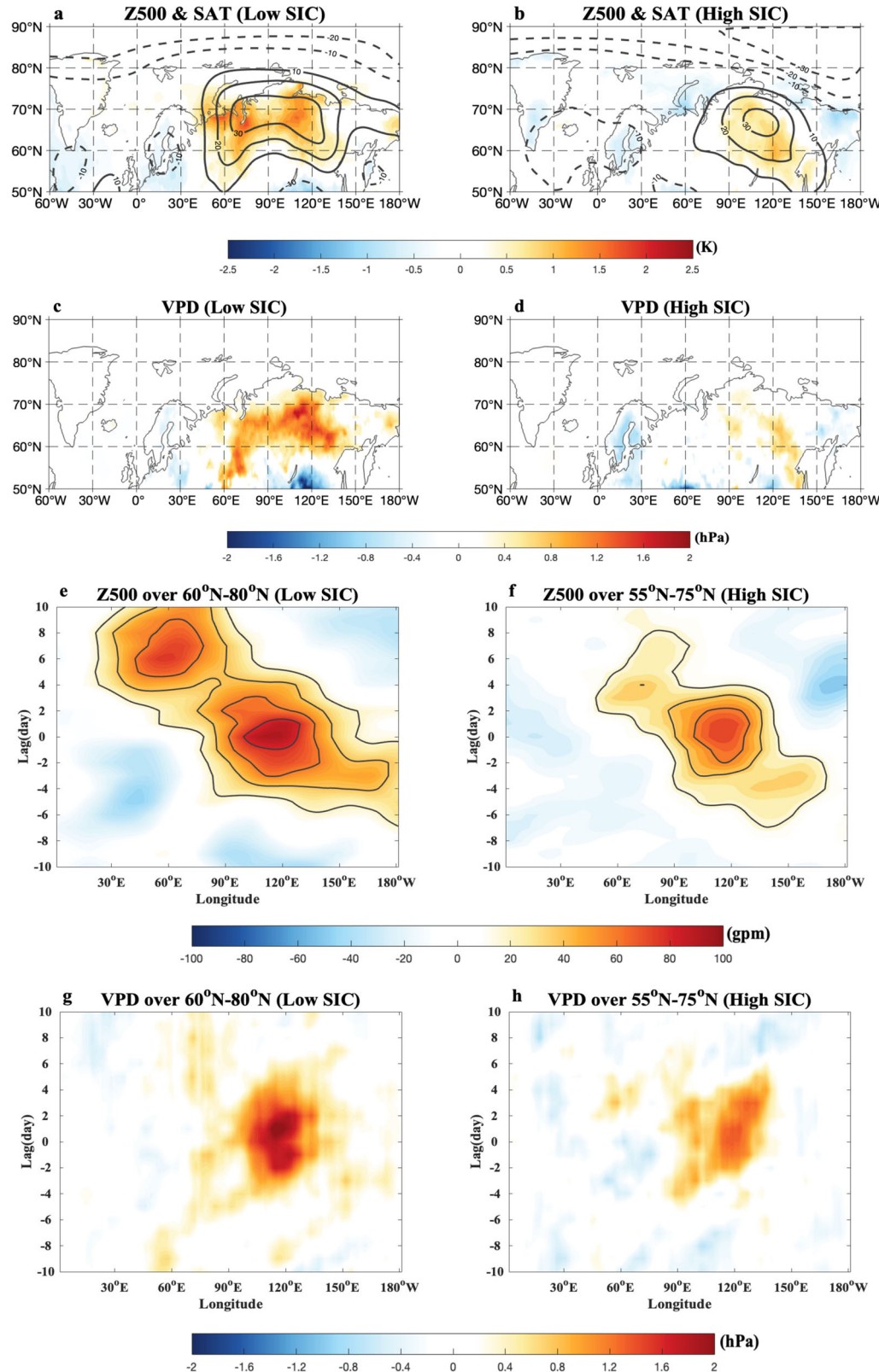

**Fig. 5 | Spatial patterns of time-mean composite daily atmospheric fields and vapor pressure deficit (VPD) anomalies averaged over the life period of Siberian blocking (SB) and time-longitude evolutions of composite daily 500-hPa geopotential height (Z500) anomalies of SB events over given latitude regions under different Arctic sea-ice conditions during 1979–2021. a–d** Time-mean fields of composite daily **a, b** surface air temperature (SAT, color shading, unit: K) and 500-hPa geopotential height (Z500, contour interval (CI) = 10 gpm) anomalies based on ERA5 data and **c, d** vapor pressure deficit (VPD; color shading, unit: hPa) anomalies averaged during the blocking life cycle from lag-10 to 10 days of Siberia blocking (SB) events for **a, c** low and **b, d** high summer (June–August, JJA) Arctic sea-ice concentration (SIC) conditions during 1979–2021, where lag 0 denotes the peak day of SB. **e–h** Time-longitude evolutions of composite daily **e, f** Z500 (CI = 10 gpm, unit: gpm) and **g, h** VPD (color shading, unit: hPa) anomalies averaged over 60°–80°N and 55°–75°N for **e, g** low and **f, h** high SIC conditions during 1979–2021. The color shading denotes the region being statistically significant above the 95% confidence level based on a two-sided student *t*-test.

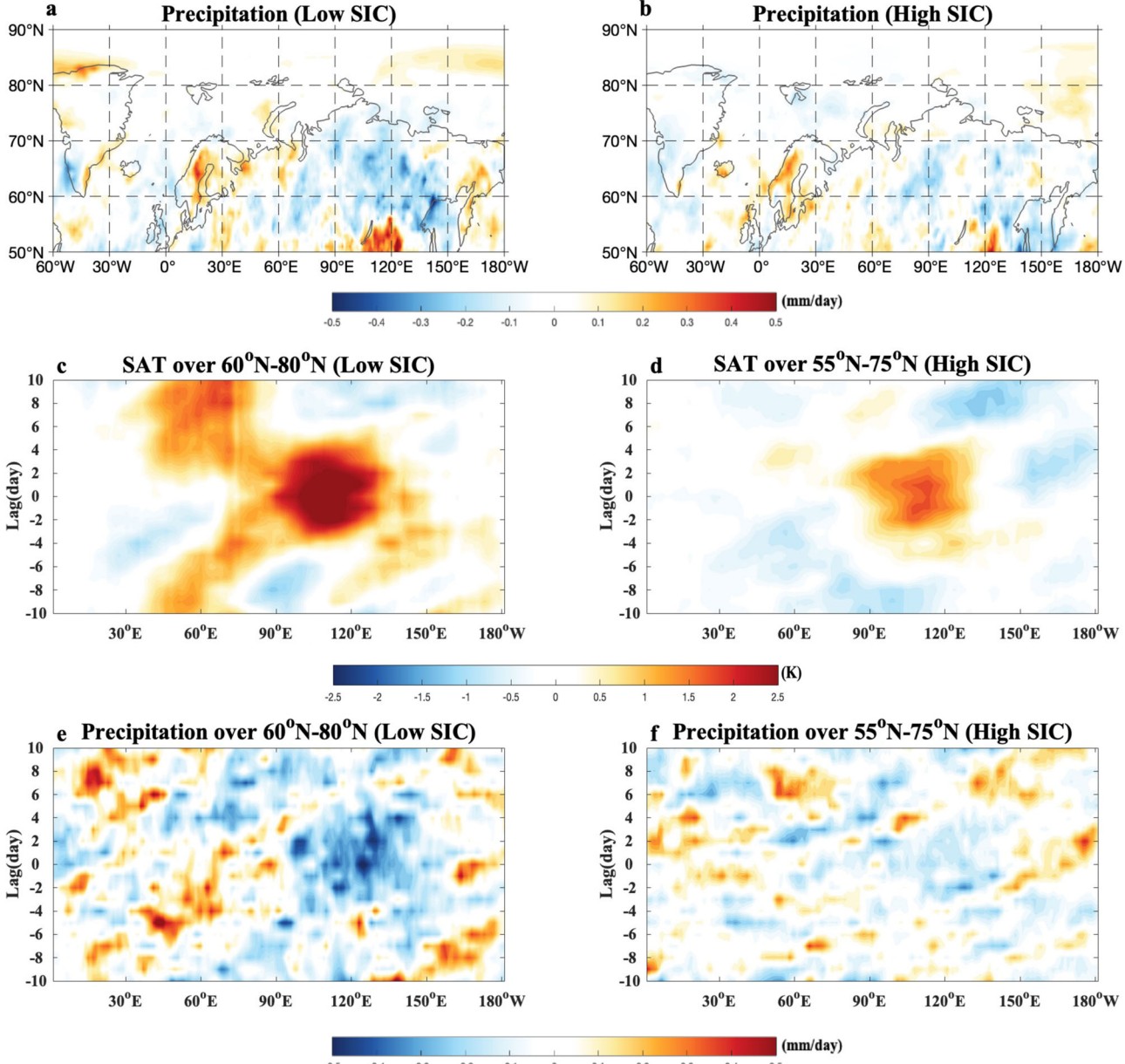

**Fig. 6 | Time-mean composite daily precipitation anomaly fields averaged over the life period of Siberian blocking (SB) and time-longitude evolution of composite daily surface air temperature (SAT) and precipitation anomalies averaged over given latitude regions during the lifecycles of SB events for different Arctic sea-ice concentration (SIC) conditions. a, b** Time-mean composite daily precipitation anomalies (unit: mm/day) averaged from lag −10 to 10 days of Siberian blocking (SB) events for **a** low and **b** high summer (June–August,

JJA) mean Russian Arctic sea-ice concentration (SIC) conditions during 1979–2021, where lag 0 denotes the peak day of SB. **c–f** Time-longitude evolution of composite daily **c, d** surface air temperature (SAT, unit: K) and **e, f** precipitation anomalies (color shading, unit: mm/day) during the life cycles of SB events averaged over 60°–80°N and 55°–75°N for **c, e** low and **d, f** high SIC conditions. The color shading denotes the region being significant at the 95% confidence level based on a two-sided student *t*-test.

more persistent and widespread surface warming, precipitation deficit and VPD increase via more persistent subsidence, short-wave radiation heating and temperature advection from lower latitudes under a low SIC condition (Figs. 5g and 6c, e) than under a high SIC condition (Figs. 5h and 6d, f). Interestingly, intense daily wildfires occur mainly during the mature and decaying phases of SB (Supplementary Fig. 10c) because the surface warming, precipitation decrease, and VPD increase are strong in these phases (Figs. 5g and 6c, e). Similar results are found for the evolution of the composite daily wildfire fraction anomaly during the blocking episode based on the GFAS data during 2003–2021 (Supplementary Fig. 11). Thus, under low SIC conditions the SB-induced wildfires can add the BAW-induced wildfires to result in stronger eastern Siberian wildfires via increased persistence, zonal scale and slow decay of SB and

its reduced eastward movement, even though the BAW associated with Russian Arctic sea-ice loss plays a major role. Similar composite results of SB events are detected for the detrended data if the eastern Siberian SAT time series is used as a proxy to define low and high Arctic warming summers (Supplementary Fig. 12).

## Mechanism of the response of Siberian blocking to enhanced eastern Siberian Arctic warming

Here we further explore why the Russian Arctic SIC loss can influence the evolution of SB via enhanced Arctic warming. In the nonlinear multi-scale interaction (NMI) model of atmospheric blocking or meandering jetstream events[68–70], the meridional background potential vorticity gradient (PV_y-PV_N−PV_S) is an important physical factor

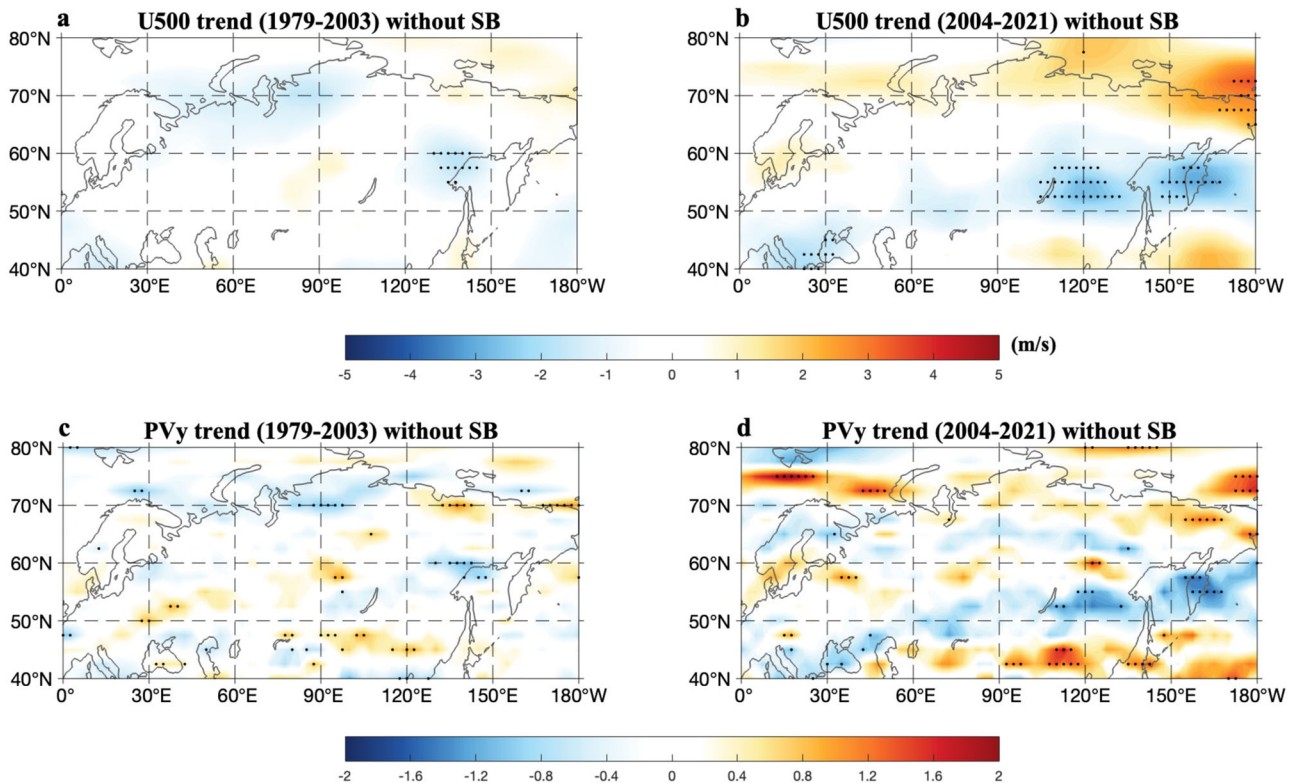

**Fig. 7 | Linear trend patterns of summer zonal wind and meridional potential vorticity gradient anomalies over 1979–2003 and 2004–2021. a–d** Linear trend patterns of summer (June–August, JJA) mean 500-hPa (**a**, **b**) zonal wind (U500) (color shading; unit: m/s per decade) and **c**, **d** non-dimensional meridional potential vorticity gradient (PV$_y$) anomalies (color shading; unit: non-dimensional value per decade) without Siberian blocking (SB) events (i.e., blocking days from lag − 10 to 10 days are removed for each SB event and lag 0 denotes the peak day of SB) over (**a**, **c**) 1979–2003 and **b**, **d** 2004–2021. The dotted regions represent that the linear trends are statistically significant ($p < 0.05$) for a two-sided student $t$-test.

characterizing the spatiotemporal evolution of atmospheric blocking, where PV$_N$ (PV$_S$) is the background potential vorticity (PV) in high (middle) latitudes. As noted above, the Arctic sea-ice loss is able to cause enhanced Arctic warming, thus it is likely that Arctic sea-ice loss influences blocking events via slow changes in PV$_y$ associated with Arctic warming[69] in that the summer sea-ice loss and associated Arctic warming have much slower timescales than blocking events. A reduced PV$_y$ appears in the south side of high latitude warming in the troposphere during 2004–2021 (Fig. 7d) because the stronger warming in high latitudes (Fig. 2f) corresponds to lower PV (PV$_N$) than in middle latitudes (PV$_S$). As a result, enhanced Arctic warming can lead to a small PV$_y$ in mid-high latitudes because of PV$_N$ < PV$_S$ as shown in a schematic diagram (Fig. 8a). Hence, the magnitude and distribution of PV$_y$ (Fig. 7) can be used to establish a bridge from the Russian Arctic SIC decline to changes in SB via the effect of BAW. According to the PV gradient theory of atmospheric blocking in the NMI model[68–70], blocks can have long lifetime, large zonal scale, weak eastward movement, and slow decay (Fig. 8a) due to weak energy dispersion and strong nonlinearity when PV$_y$ is small[68–70]. Such blocking changes can cause strong, widespread, and long-lasting heatwaves (Fig. 8a), thus promoting intense and persistent wildfire events in large areas. The mathematical form of PV$_y$ and its numerical calculation and physical meaning are briefly described in "Methods" section.

It is noted that the low (high) SIC corresponds to a small (large) PV$_y$ in the eastern Siberian region (50°–65°N, 90°–150°E) from the Ural Mountains to eastern Siberia (Supplementary Fig. 13), thus favoring (inhibiting) long lifetime, large zonal scale, weak eastward movement and slow decay of SB (Fig. 5e, f) to promote (suppress) intense, widespread and persistent heatwave events over eastern Siberia (Fig. 6c, d). We further see that 2004–2021 corresponds to large negative

anomalous background zonal wind (Fig. 7b) and PV$_y$ (Fig. 7d) trends over eastern Siberia because the BAW trend is strong (Fig. 2f) due to strong Russian Arctic SIC loss (Fig. 1f). In contrast, 1979–2003 corresponds to small negative background zonal wind (Fig. 7a) and PV$_y$ (Fig. 7c) trend anomalies because the BAW trend is weak. Thus, the recent accelerated decline of Russian Arctic SIC leads to strong negative background zonal wind[55,56] and PV$_y$ trend anomalies in the latitude belt (50°–65°N) of eastern Siberia via enhanced BAW which favors the increasing trends in the lifetime, zonal scale, weak eastward movement and slow decay of SB, factors which are conducive to increases in eastern Siberian wildfires in lifetime and range during 2004–2021. In brief, the rapid loss of Russian Arctic SIC as observed during 2004–2021 tends to increase the persistence, zonal scale, westward movement and slow decay of SB events to result in SB-induced increases in the trend and range of wildfires over eastern Siberian high latitudes through reducing PV$_y$ over eastern Siberia due to the effect of the slowly varying BAW.

## Discussion
In previous studies[39,40,57], VPD has been shown to be a key meteorological factor that influences fire activity in boreal forests. Thus, the linear slope of the JJA-mean VPD over 2004–2021 can be used to represent the recent trend of eastern Siberian wildfires. Based on an approach ("Methods" section), in this study we use daily-mean SAT and VPD to make an estimate that ∼79% of the increasing trend of summer VPD over 2004–2021 directly results from enhanced BAW due to the Russian Arctic sea-ice decline, whereas ∼21% of the upward VPD trend comes from atmospheric internal variability via decadal changes in SB events induced by the BAW. Thus, the enhanced BAW associated with the summer Russian Arctic sea-ice decline over 2004–2021 plays a

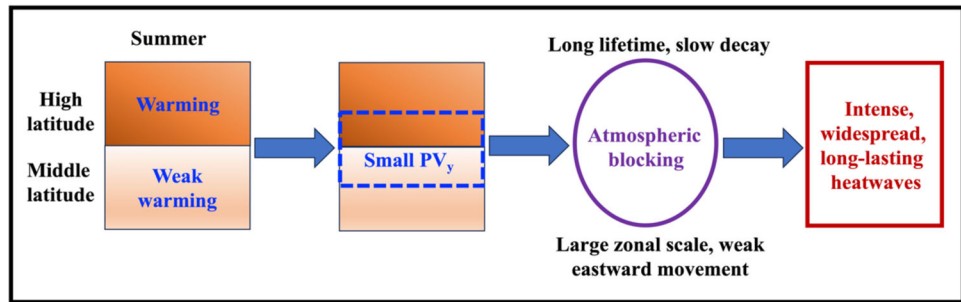

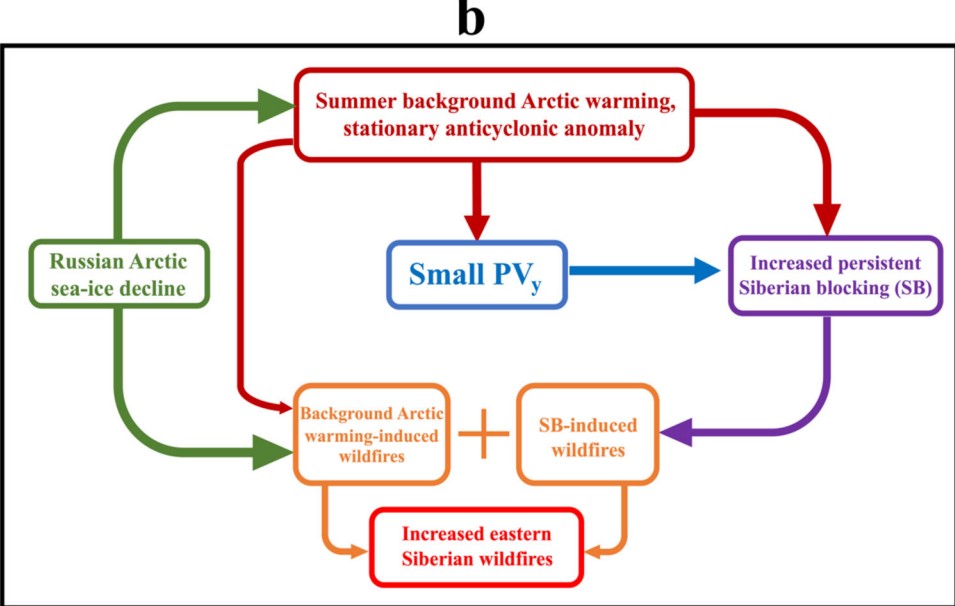

**Fig. 8 | Schematic diagrams of the physical mechanisms of summer Arctic warming influencing atmospheric blocking and summer Russian Arctic sea-ice decline affecting eastern Siberian wildfires via changes in background Arctic warming, stationary anticyclonic anomaly, and Siberian blocking (SB) events. a** Schematic diagram of the influence of summer Arctic warming on atmospheric blocking via reducing meridional background potential vorticity gradient ($PV_y$), in which a small $PV_y$ favors increased persistence, zonal scale, westward movement, and slow decay of atmospheric blocking and **b** the pathway of the influence of Russian Arctic sea-ice loss on eastern Siberian wildfires via Arctic warming is described as follows: Russian Arctic sea-ice decline (green box) can induce background Arctic warming (BAW) and stationary anticyclonic anomaly (heavy red box) over eastern Siberia, leading to a small $PV_y$ (blue box) in the south side of eastern Siberian high latitudes, which induces longer-lasting Siberian blocking (SB) events with larger zonal scale, less eastward movement and slower decay (purple box). The joint role of the background Arctic warming- and SB-induced wildfires (orange box) leads to increased eastern Siberian wildfires in high latitudes (coral red box).

major role in the recent increasing trend of eastern Siberian high latitude wildfires due to decreased precipitation, while the increased persistence, zonal scale, westward movement and slow decay of SB events favor intense, persistent and widespread wildfire events over the region. One can see an intensified connection of FWI or VPD to the SAT variability over eastern Siberia when the Russian Arctic SIC decline is strong. This indicates that the rapid sea-ice loss can significantly strengthen the influence of eastern Siberian SAT on eastern Siberian wildfires over 2004–2021. Thus, a rapid loss of summer Russian Arctic sea-ice can greatly enhance the risk of recent eastern Siberian high-latitude wildfires.

As shown in Fig. 8b, our results indicate that the strong SIC loss leads to strong BAW and stationary anticyclonic anomaly in the eastern Siberian high latitudes (Fig. 2f), which provide favorable climatic conditions for increased wildfire ignition via increased Arctic lightning[20,57], decreased precipitation (Fig. 3c), reduced soil moisture[20,22] and increased VPD over eastern Siberia and then dominate the major trend of eastern Siberian wildfires. Moreover, strong BAW and anticyclonic anomaly can also produce a small $PV_y$, which is conducive to the high latitude SB and its increased persistence, zonal scale, westward

movement and slow decay. Such blocking changes can lead to intense, widespread and persistent wildfire events over eastern Siberian high latitudes (Supplementary Figs. 10 and 11) via generating heatwaves[66,67,71] and high VPD (Fig. 5) during the blocking episode, thus amplifying the increasing trend of summer eastern Siberian high-latitude wildfires over 2004–2021 caused by the BAW trend associated with the Russian Arctic SIC loss. Clearly, the BAW also plays an indirect role through influencing SB events in addition to its direct role. Therefore, Arctic amplification, Russian Arctic sea-ice loss and atmospheric internal variability (e.g., long-term changes in SB events) jointly dominate the increasing trend of eastern Siberian high-latitude wildfires over 2004–2021 (Fig. 8b). In fact, enhanced Arctic warming also appears in spring, autumn and winter[33] due to increasing $CO_2$ and sea-ice loss[34]. Thus, atmospheric blocking in a smaller $PV_y$ region will become more persistent and have larger zonal scale, less eastward movement and slower decay so that more intense, widespread and persistent wildfire events also likely occur in spring and autumn via heatwaves (Fig. 8a) if $PV_y$ remains smaller even under the future Arctic warming situation or in a future warmer climate.

We remind the reader that our study does not discuss the interannual variability of the eastern Siberian wildfires and the potential

role of El Niño-Southern Oscillation (ENSO) while noting that ENSO has the impact on Russian blocking activity[72]. Our focus here has been on the recent trend of wildfires, although the AMO and PDO are considered as climatic factors influencing the slow variations of Russian Arctic SIC[37] and slow changes in SB events via Arctic warming. Since eastern Siberian wildfires result from multiple weather and climate factors as well as their multi-scale interaction processes, separating the different contributions of these factors and processes in the Siberian wildfire variability and trend is still a big challenge.

In this study, although we have estimated the relative contributions of enhanced BAW and internal atmospheric variability to the recent increasing trend of eastern Siberian wildfires over 2004–2021 by calculating VPD, it is difficult to separate individual contributions of four meteorological variables: air temperature, relative humidity, wind speed and precipitation to eastern Siberian wildfires. This is because the four variables are not independent of each other and also because the relative humidity, wind speed and precipitation depend on air temperature. For this reason, we use VPD instead of the four meteorological variables to examine the cause of the increases in recent high-latitude wildfires over eastern Siberia. Moreover, the tree species also affect boreal wildfires[73]. In particular, different tree species over Siberia and North America can cause a large difference of wildfires between the two continents in intensity, burned area, and frequency. For example, wildfires in the larch-dominated forests of Siberia result in high tree mortality and carbon loss[74].

On the other hand, the effects of other factors such as wildfire feedback, snowmelt, vegetation coverage, peatland burning, frozen soil and hydrological cycle are not considered in our study[75,76], even though the wildfire-induced $CO_2$ emissions feedback can cause an increase of about 0.18 °C in the land SAT[77]. Previous studies also noted that in recent decades the increasing trends of summer wildfires and associated $CO_2$ emissions are stronger over eastern Siberia than over North America[17]. This difference may be related to a fact that the summer SIC loss is more prominent over the Russian Arctic side than over North American Arctic side. We therefore suggest that these issues should be further investigated using integrated satellite data, future global climate models that can realistically simulate blocking and wildfire events, and machine learning-based modeling approaches, especially from atmospheric blocking perspectives. Nevertheless, our study provides an attempt to identify the relative contributions of the Russian Arctic sea-ice decline and internal atmospheric variability associated with SB events to the recent increasing trend of eastern Siberian high latitude wildfires, while wildfires in Northeast China and North America have been shown to be linked to Arctic sea-ice declines in the Bering Sea region[78,79].

## Methods

### Definition and physical meaning of vapor pressure deficit

Vapor pressure deficit (VPD) is defined as the difference between the saturation vapor pressure ($e_s$) and actual vapor pressure ($e_a$), which is written as[80]

$$VPD = e_s - e_a \qquad (1)$$

where $e_s = 6.109 \times e^{\left(\frac{17.625 \times T}{T + 243.04}\right)}$ and $e_a = 6.109 \times e^{\left(\frac{17.625 \times T_d}{T_d + 243.04}\right)}$, where $T$ is the near surface (2 m) air temperature (°C) and $T_d$ is the dew point temperature (°C).

VPD is the combination of air temperature and relative humidity[62]. When there are high air temperature and low relative humidity, VPD is high so that it creates the most fire-prone conditions and increases fire ignition, growth and burned area by dying of fuel[39,80]. High VPD is often linked to drought or low soil moisture and precipitation deficit[81] and related to winds associated with high pressure systems[40]. High VPD conditions have been shown to reduce stomatal conductance and photosynthesis and to increase plant mortality while simultaneously

increasing plant water losses through transpiration[82]. Thus, the value of VPD is often used as a measure of wildfire risk[40,62].

The sensitivity of monthly-mean VPD to the monthly mean values of daily maximum and minimum air temperature ($T_{max}$ and $T_{min}$) has been examined and a small error was found in a previous study[39], even though daily $T_{max}$ is used in some studies[7,57,58] and the nighttime air temperature (or $T_{min}$) is used to compute nighttime VPD in some studies[62,80]. Here, VPD can explain more of the variance in fire activity than can precipitation, drought indices, air temperature, or wind individually, and is more successful in explaining the ignition, spread, intensity and size of forest fires than other meteorological variables[39]. Thus, in this study we use the value of VPD to represent the risk of eastern Siberian wildfires. In previous studies[55,56,66,67], the daily mean air temperature and its monthly mean data were often used to calculate Arctic warming or Arctic amplification in different seasons[33]. In this study, we use the daily mean SAT rather than daily $T_{max}$ to compute the Arctic warming and VPD over eastern Siberia, even though the effect of daily $T_{max}$ is also discussed.

### Fire weather index and its physical meaning

The Fire weather index (FWI) is a meteorologically based index used worldwide to estimate fire danger. The Global ECMWF Fire Forecast model is used as a fire danger model to produce the FWI data, which includes the effect of air temperature, relative humidity, wind speed, precipitation, drought conditions, fuel availability and vegetation characteristics by considering daily noon values of air temperature (or $T_{max}$), relative humidity, wind speed and 24-h accumulated precipitation.

The FWI indicates fire intensity by combining the rate of fire spread with the amount of fuel being consumed. Moreover, the fire intensity can be further categorized in terms of the value of FWI under a changing climate.

### Blocking index and identification of Siberian blocking events

To identify Siberian blocking events in the region 90°–120°E, we used the one-dimensional blocking index of Tibaldi and Molteni (TM hereafter)[83]. The TM index is constructed based on the reversal of the meridional 500-hPa geopotential height (Z500) gradient at a given time:

$$GHGN = \{Z500(\phi_N) - Z500(\phi_o)\}/\{\phi_N - \phi_o\} \qquad (2)$$

$$GHGS = \{Z500(\phi_o) - Z500(\phi_S)\}/\{\phi_o - \phi_S\} \qquad (3)$$

at three given latitudes $\phi_N = 80^o N + \Delta$, $\phi_o = 60^o N + \Delta$, $\phi_S = 40^o N + \Delta$ and $\Delta = -4^o, 0^o, 4^o$. The results of using $\Delta = -5^o, 0^o, 5^o$ also have no large difference with those of using $\Delta = -4^o, 0^o, 4^o$. A blocking event is defined to have taken place if the conditions GHGS > 0 and GHGN < −10 gpm (deg lat)⁻¹ hold for at least three consecutive days in a prescribed zonal region covering at least 15° of longitude. In this study, the region (90°–150°E, 60°–75°N) is referred to as the eastern Siberia, even though Siberian blocking events are picked up in the region 90°–120°E.

### Calculation of ocean heat transport

The zonal ocean heat transport (OHT) across the latitude-depth cross-section at longitude $x$ and a given time is given by

$$OHT(x) = \rho C_p \int_{y_1}^{y_2} \int_z^0 u(x,y,z)\theta(x,y,z)dzdy \qquad (4)$$

where $\rho$ is the density (assumed constant), $C_p$ is the heat capacity of seawater, and z is the upper ocean depth which is set as 150 m. Also, $y_1$ and $y_2$ are the southern and northern latitudes of the cross-section, $\theta$ is the monthly ocean potential temperature, and $u$ is the monthly mean zonal velocity[84,85].

## CESM1 simulations

We used the fully coupled Community Earth System Model version 1 (CESM1) with a horizontal resolution of 2.5° longitude × 2.0° latitude for the atmospheric model, and 1.0° longitude × 0.5° latitude for the sea-ice and ocean models, which is available from the National Center for Atmospheric Research. This CESM1, which well simulates the Arctic SIC and climate, is used to perform 235-year simulations plus a 150-year pre-industrial control run. Here, two simulation experiments with 1%-per-year increase in atmospheric carbon dioxide ($CO_2$) with and without fixed sea-ice conditions are performed and used to examine whether enhanced summer Arctic warming is caused by the sea-ice loss under a constant external forcing. The details of the CESM1 simulation design can be found in the reference[34]. The difference between the 1% $CO_2$ run with fully interactive sea ice and the 1% $CO_2$ run with fixed sea-ice clearly reveals that the summer (JJA) sea-ice loss can cause stronger summer Arctic warming over the Eurasian Arctic than the fixed sea-ice.

## The Polar Amplification Model Intercomparison Project (PAMIP) ensemble of CMIP6 simulations

We used two PAMIP ensembles forced by future and pre-industrial Arctic sea-ice concentrations (SICs) to examine how the SIC forcing field leads to Arctic warming in summer. Each PAMIP ensemble consists of 1000 runs from 10 models of CMIP6 which include AWI-CM-1-1-MR, CESM2, CanESM5, CESM1-WACCM-SC, CNRM-CM6-1, IPSL-CM6A-LR, NorESM2-LM, FGOALS-f3-L, MIROC6 and HadGEM3-GC31-MM. The given future and pre-industrial SIC forcing fields can be found in ref. 48.

## Data treatment and statistical significance tests

When performing analyses and composites for atmospheric variable, sea-ice and fire fields, the de-seasonalized daily and monthly anomaly fields can be obtained by removing the long-term (1979–2021) mean of all the daily or monthly data for each calendar day or month. We also used a two-sided student $t$-test to test the statistical significance of the composite anomaly and difference fields[86]. The significance of the correlation and the slope rates of the linear trends during the different sub-periods can be tested by using the student's $t$-test or Mann–Kendall (MK) test[86]. The 95% confidence level is denoted by $p < 0.05$.

## The method of calculating the time series of wildfire, Arctic warming, and sea-ice and their linear trend patterns

We examined trends in the time series of the FWI and SAT anomaly averaged over eastern Siberia (90°–150ºE; 60°–75ºN) and sea-ice concentration (SIC) anomaly averaged over the Russian Arctic (30°–130ºE; 65°–85ºN). The spatial pattern of the linear trend for a variable (in our case, SAT or Z500 or SIC) can be further derived by calculating its trend value at each grid point. Then, the statistical significance of the trend can be performed by using the student's $t$-test or MK test.

## Cause-effect relationship between Siberian blocking or wildfire events and Arctic warming over eastern Siberia linked to Russian Arctic SIC decline

Although our correlation analysis cannot directly establish the cause-and-effect relationship between the eastern Siberian Arctic warming associated with Russian Arctic SIC decline and Siberian blocking or wildfire events, the simulation results reveal that the increase in eastern Siberian Arctic warming is mainly caused by the sea-ice loss even if sub-seasonal atmospheric circulation patterns (e.g., Siberian blocking) are absent. Such an enhanced Arctic warming or Arctic amplification represents the background Arctic warming trend.

While the Siberian blocking event is generally of sub-seasonal timescales (≤30 days) and much shorter than that of the BAW, the long-time change in Siberian blocking events is believed to primarily result from the slow modulation of the BAW due to the slow variability of Russian Arctic SIC[67,71] because atmospheric blocking events can be modulated by the slowly varying BAW via slow changes in the meridional potential vorticity gradient according to the blocking theory[68–70] as presented below. In contrast, the impact of individual Siberian blocking events on the slow variation of the BAW and Russian Arctic SIC is weak.

On the other hand, because SB events with long lifetimes, large zonal scale, weak eastward movement and slow decay can produce intense, persistent, and widespread wildfire events (≤30 days), the long-time change in Siberian blocking events modulated by the long-time variation of the BAW due to the long-time varying forcing of Russian Arctic SIC caused by AMO and PDO can induce a long-time variation of eastern Siberian wildfire events. For example, a low (high) SIC condition favors a long (short) lifetime, large (small) zonal scale, weak (strong) eastward movement, and slow (fast) decay of high (middle) latitude Siberian blocking. Thus, the increases in the lifetime, zonal scale, westward movement and slow decay of these SB events from a high SIC state to a low SIC state can lead to increases in eastern Siberian high-latitude wildfire events in strength, range, and persistence. This provides an explanation for the role of the Russian Arctic SIC decline in changes in blocking and wildfire events.

Because the enhanced BAW corresponds to intensified land-surface warming and decreased precipitations over eastern Siberia, the BAW trend associated with the SIC loss mainly determines the overall trend of eastern Siberian wildfires over 2004-2021. This is the direct role the BAW trend plays. However, the enhanced BAW trend leads to an additional increasing trend in eastern Siberian wildfire events via increases in the persistence, zonal scale, westward movement, and slow decay of Siberian blocking events. This is an indirect role the BAW trend plays via internal atmospheric variability. Thus, the combined effect of the enhanced BAW trend and internal atmospheric variability associated with slow changes in Siberian blocking events can cause significant increases in eastern Siberian high-latitude wildfires over 2004–2021.

## An estimate for the contributions of the background Arctic warming and Siberian blocking events to the increasing trends of the surface air temperature and vapor pressure deficit over eastern Siberia

In this study, a method is presented to estimate the contributions of the BAW trend and changes in Siberian blocking events to the increasing trend of eastern Siberian surface air temperature (SAT). In this method, we assume that during 2004–2021 the increasing trend of the summer eastern Siberian Arctic SAT time series can be expressed in the form of

$$T_{SAT} = T_{BAW} + T_{SB} \tag{5}$$

Here, $T_{SAT}$ represents the JJA-mean eastern Siberian Arctic SAT anomaly during 2004–2021, $T_{BAW}$ is the background eastern Siberian Arctic warming (BAW) or SAT anomaly mainly associated with the Russian Arctic SIC decline during 2004–2021 for the case without Siberian blocking events and $T_{SB}$ denotes the JJA-mean eastern Siberian Arctic SAT anomaly only related to SB events during 2004–2021. Then, the slope rate of the eastern Siberian Arctic SAT anomaly during 2004–2021 can be expressed as

$$S_{SAT} = S_{BAW} + S_{SB} \tag{6}$$

where $S_{SAT} = \partial T_{SAT}/\partial t$, $S_{BAW} = \partial T_{BAW}/\partial t$ represents the linear slope of the BAW or its trend over 2004–2021 and $S_{SB} = \partial T_{SB}/\partial t$ represents the linear trend of the eastern Siberian Arctic SAT anomaly caused by changes in Siberian blocking events or the SB-induced SAT trend over the same period.

One can have $S_{SB} \approx 0$ and $S_{SAT} \approx S_{BAW}$ if Siberian blocking events are removed. In this case, the values of $S_{SAT}$ and $S_{BAW}$ can be obtained by calculating the slope rates of the JJA-mean eastern Siberian Arctic SAT time series with and without Siberian blocking events. According to the values of $S_{SAT}$ and $S_{BAW}$, one can estimate the ratio $R_{SB} = S_{SB}/S_{SAT}$ produced by the internal atmospheric variability associated with changes in Siberian blocking events in terms of

$$1 = R_{BAW} + R_{SB} \tag{7}$$

where $R_{BAW} = S_{BAW}/S_{SAT}$ represents the contribution from the BAW trend. By comparing the values of $R_{BAW}$ and $R_{SB}$, we can estimate the relative contributions of the BAW trend associated with the Russian Arctic SIC loss and internal atmospheric variability (i.e., changes in SB events) to the increasing trend of the recent summer eastern Siberian SAT. If the vapor pressure deficit (VPD) is used instead of SAT, one can estimate the relative contributions of the BAW and internal atmospheric variability to the increasing trend of the VPD that controls the increase in eastern Siberian wildfires. Because the climate models cannot realistically simulate internal atmospheric variability (e.g., Siberian blocking events), using the climate models to estimate the different contributions from the Russian Arctic SIC decline and internal atmospheric variability to the recent trend of Siberian wildfires is difficult. However, our approach used here can better resolve this issue.

**Potential vorticity gradient theory of atmospheric blocking and the calculation of the meridional background potential vorticity gradient (PV$_y$) and the physical mechanism of Arctic sea-ice loss influencing atmospheric blocking**

In the nonlinear multiscale interaction (NMI) model of atmospheric blocking (i.e., meandering jet-stream) events[68–70], the meridional background potential vorticity gradient (PV$_y$) is shown to be a key factor that influences the spatiotemporal evolution of atmospheric blocking[68]. When PV$_y$ is smaller, the blocking system has weaker non-dispersion and stronger nonlinearity to allow atmospheric blocking to have longer lifetime, less eastward movement, larger zonal scale, and slower decay[68,69]. When PV$_y$ is slowly varying in time, the blocking event will show a slow variation. Here, we used the non-dimensional three-dimensional baroclinic PV$_y$ scaled by the characteristic velocity $\widetilde{U}$ (-10 m/s), length $\widetilde{L}$ (-10$^6$ m), and horizontal height $\widetilde{H}$ (-10$^4$ m). The non-dimensional PV$_y$ can be expressed as

$$\mathrm{PV}_y = \beta - \frac{\partial^2 U}{\partial y^2} - \frac{1}{\rho_0}\frac{\partial}{\partial z}\left(\rho_0 F_r \frac{\partial U}{\partial z}\right) \tag{8}$$

where $\rho_0 = \rho_s e^{-z}$ with $\rho_s$ being the density of atmospheric reference state at the Earth's surface ($z = 0$) and $F_r = \frac{\widetilde{L}^2 f_0^2}{N^2 \widetilde{H}^2}$, where $\beta = \beta_0 \widetilde{L}^2 /\widetilde{U}$, $N^2$ is the Brunt-Vaisala frequency, $U(y, z)$ is the non-dimensional basic zonal wind, $f_0$ and $\beta_0$ are the Coriolis parameter and its meridional gradient at a given reference latitude $\varphi_0$, respectively. We set $F_r = 0.72$ as a fixed parameter[70] and $z = 1$ as the tropopause.

The physical mechanism of how the Arctic sea-ice loss influences atmospheric blocking has been in detail discussed in previous studies[69]. The Arctic sea-ice loss can induce stationary anticyclonic and warm anomalies over the Arctic high latitudes because the solar radiative heating warms the tropospheric atmosphere and induces a positive height anomaly as the sea-ice is melting[54]. As a result, PV$_y$ is reduced in the south side of the high-latitude stationary anticyclonic or warm anomalies. This small PV$_y$ can increase the persistence, zonal scale, westward movement and slow decay of atmospheric blocking via reducing energy dispersion and strengthening the nonlinearity of the blocking system[69].

## Data availability

We used the monthly and daily surface air temperature (SAT) at 2 m, 500-hPa geopotential height (Z500), 500-hPa zonal wind (U500) component fields with 1°×1° grid points in summer (June to August, JJA) from December 1979/February 1980 to December 2021/February 2022 (1979-2021), which were obtained from the European Centre for Medium-range Weather Forecasts ERA5 reanalysis dataset[51] (https://climate.copernicus.eu/climate-reanalysis). The summer monthly mean sea ice concentration (SIC) data are taken from the Hadley Centre surface temperature (HadISST) dataset with 1°×1° grid points (https://www.metoffice.gov.uk/hadobs/index.html).

The summer PDO index we used here is taken from the Koninklijk Nederlands Meteorologisch Instituut (KNMI) Climate Explorer (http://climexp.knmi.nl/getindices.cgi?WMO=UWData/pdo_hadsst3&STATION=PDO_HadSST3&TYPE=i&id=someone@somewhere), whereas the summer AMO index is available from the Koninklijk Nederlands Meteorologisch Instituut (KNMI) Climate Explorer (http://climexp.knmi.nl/getindices.cgi?WMO=UKMOData/amo_hadsst&STATION=AMO_hadsst&TYPE=i&id=someone@somewhere).

We downloaded the summer daily and monthly Fire Weather Index (FWI) data based on the fire danger indices historical data with 0.25°×0.25° grid points from the Copernicus Emergency Management Service for the European Forest Fire Information System (EFFIS) for the period 1979- 2021 (https://cds.climate.copernicus.eu/cdsapp#!/dataset/10.24381/cds.0e89c522?tab=overview). The EFFIS incorporates the fire danger indices for three different models developed in Canada, United States, and Australia, but not parameterized only based on Canadian fire.

The daily wildfire fraction used in this study was derived from the Global Fire Assimilation System (GFAS) with horizontal resolution of 0.1°×0.1° grid points from 2003 to 2021 based the satellite observation (https://www.ecmwf.int/en/forecasts/dataset/global-fire-assimilation-system). We also used the monthly mean burned area data from the MODIS Fire Climate Change Initiative (Fire CCI) version 5.1 products (FireCCI51) with horizontal resolution of 0.5°×0.5° grid points from 2001 to 2020 based on the satellite observation (https://data.ceda.ac.uk/neodc/esacci/fire/data/burned_area/MODIS/grid/v5.1).

The monthly burned fraction data used in this study was taken from Global Fire Emissions Database (GFED), Version 4 with horizontal resolution of 0.25°×0.25° grid points from 1997 to 2016 combined with the satellite observation (Global Fire Emissions Database (globalfiredata.org)).

We also used monthly burned area data on a 0.25° × 0.25° degree resolution grid from 1982 to 2018 derived from satellite observations and from spectral information from the AVHRR (Advanced Very High Resolution Radiometer) Land Long Term Data Record (LTDR) v5 dataset produced by NASA, which is referred to as AVHRR-LTDR (https://catalogue.ceda.ac.uk/uuid/62866635ab074e07b93f17fbf87a2c1a).

The JJA-mean SAT data under future and pre-industrial Arctic SICs used to perform the Polar Amplification Model Intercomparison Project (PAMIP) ensemble are taken from 10 models of CMIP6.

## Code availability

The code of the CESM1 model used here is available from (http://www.cesm.ucar.edu/models/cesm1.2/). Other codes used in the present study are available from the corresponding author on request.

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

## Acknowledgements

This research was supported by the National Natural Science Foundation of China (Grant number: 42150204, 42288101), the National Key Research and Development Program of China (2023YFF0805100), and the Beijing Normal University Talent Introduction Project of China (12807-312232101, 2022-GJTD-01). B.L. was supported by the China National Postdoctoral Program for Innovative Talents (BX20230045) and the China Postdoctoral Science Foundation (2023M730279). I.S. was supported by the Australian Research Council (Grant DP 160101997).

## Author contributions

B.L. performed this study, plotted all figures in the main text and a large proportion of supplementary figures, and wrote the preliminary manuscript. D.L. supervised this study and revised the preliminary manuscript in part. A.D., C.X., I.S., E.H., J.O., J.S., X.C., Y.Y., W.D., Y.L., Q.Z., X.X., Y.D., Z.J., and T.G. joined this project and gave some discussions on the improvement of this manuscript. All the authors contributed to the writing and reviewing of the manuscript.

## Competing interests

The authors declare that they have no competing interests.
