## [Peer Review File · Nature Communications]

Rapid summer Russian Arctic sea-ice loss enhances the risk of recent Eastern Siberian wildfiresEditorial Note: Parts of this Peer Review File have been redacted as indicated to remove third-party material where no permission to publish could be obtained.

REVIEWER COMMENTS

Reviewer #1 (Remarks to the Author):

Luo et al. 2023 illustrated the impact of the declining trend of sea ice on Eastern Siberian fires. They analyzed atmospheric circulation blocking events, and their variation with time contributes to stronger recent fire-prone weather and frequent fire events. I agree that this hypothesis works in the real world, but I think they did not provide enough pieces of evidence in detail:

1. Correlation cannot prove cause-and-effect relationship.

Authors show interannual correlations between JJA-mean sea ice cover (SIC) and the Fire Weather Index (FWI) and also for detrended data. While correlation values for -0.37 and -0.27 are significant at a 95% confidence level, they cannot prove cause-and-effect relationship. For example, temperature (known as Arctic amplification) also leads to sea ice decline and fire, so it cannot be separated into the sole role of sea ice on fire and blocking.

With trends, temperature, and sea ice might have the same pattern with time, so it would be easy just to use temperature, and I am not sure why authors focus on sea ice instead of just temperature because the temperature is one of the components for calculating FWI.

2. Fire data

There is no direct observation of the Siberian region's fire, but it would be better to use remotely sensed fire activity directly instead of FWI because FWI is parameterized based on Canadian fire, not Siberia. As the authors already used GFAS data, I think the main results should be conducted with GFAS data, such as the correlation between GFAS and sea ice. Also, many remote-sensed fire datasets, such as MCD64, GFED, and Fire Radiative Power, are available to compare reality. In addition, a previous study by Rogers et al. 2015, pointed out that fire characteristics in Siberia and North America are distinctly different, so FWI is unsuitable for Siberian fire studies.

Rogers, B., Soja, A., Goulden, M. et al. Influence of tree species on continental differences in boreal fires and climate feedbacks. *Nature Geosci* 8, 228–234 (2015). <https://doi.org/10.1038/ngeo2352>

3. Global climate (Earth system) model simulation is needed.

If authors want to provide solid evidence of sea ice's role in fire and blocking, I think authors need to run climate model (Earth system model) simulations to provide differences between sea ice decline simulations and control simulations. Then, it clearly finds how sea ice loss occurs by blocking and fire-prone weather, but only correlation analysis cannot provide enough background.

Reviewer #2 (Remarks to the Author):

Please see the attachment.

[Editorial Note: This is displayed on the next page]

Review of Luo et al.

Base on the meteorological and satellite data, this study has revealed most of the surge in eastern Siberian wildfires from 2004 to 2021 is attributable to the background Arctic warming linked to the decline in Russian Arctic sea ice. The remaining is associated with internal atmospheric variability linked to changes in Siberian blocking events. Furthermore, over the period 2004-2021, Siberian blocking events at higher latitudes exhibit increased persistence and larger zonal scales compared to the period 1979-2003, attributed to reduced meridional potential vorticity gradients resulting from background Arctic warming. This topic is interesting and highly original. The author has provided a new perspective to understand the recent increases in eastern Siberian wildfires. I thought that this manuscript has potential to be published in Nature Communication. Thus, I recommend that the manuscript needs minor revision.

1, This article, in my view, places more emphasis on revealing the connection between sea ice, blocking, and wildfires with high temperatures and circulation patterns. In fact, high temperature is one of the most important elements. However, based on the FWI, besides of temperature, the relative humidity, precipitation and wind can be also contributed to the variations of FWI. It lacks a thorough demonstration of the specific processes leading to wildfires, relying heavily on literature references. I suggest that authors can add some discussions about the impacts and contributions of these potential factors.

2, Authors have mentioned AMO, PDO, blockings, with different time scales. As investigating the causes of wildfires, particularly as extreme events, special attention should be given to the issue of matching time scales.

3, Line 92, authors mentioned using a novel approach. Please clarify the “new” in the approach.

4, line 215 How to understand 84%, a very high percentage?

Response to Reviewer #1's comments for the manuscript (NCOMMS-23- 54142)

The authors would like to thank Reviewer #1 for his/her constructive comments. We have made substantial revisions according to your remarks (in italic text). Our point-by-point responses (in plain text) are given below:

Luo et al. 2023 illustrated the impact of the declining trend of sea ice on Eastern Siberian fires. They analyzed atmospheric circulation blocking events, and their variation with time contributes to stronger recent fire-prone weather and frequent fire events. I agree that this hypothesis works in the real world, but I think they did not provide enough pieces of evidence in detail:

Response:

We appreciate your useful overview here. We have made a major revision based on your comments.

Response to Major comments:

Question:

1. Correlation cannot prove cause-and-effect relationship.

Authors show interannual correlations between JJA-mean sea ice cover (SIC) and the Fire Weather Index (FWI) and also for detrended data. While correlation values for -0.37 and -0.27 are significant at a 95% confidence level, they cannot prove cause-and-effect relationship. For example, temperature (known as Arctic amplification) also leads to sea ice decline and fire, so it cannot be separated into the sole role of sea ice on fire and blocking. With trends, temperature, and sea ice might have the same pattern with time, so it would be easy just to use temperature, and I am not sure why authors focus on sea ice instead of just temperature because the temperature is one of the components for calculating FWI.

Response:

We agree that the correlation cannot reveal a cause-effect relationship. In our study, we don't use the correlation to establish the cause-effect relationship between two variables (A and B). Rather, we use the magnitude of the correlation coefficient to judge whether two variables potentially have some association. A stronger correlation coefficient between FWI index over eastern Siberia and Russian Arctic SIC over 2004-2021 indicates that the connection of the eastern Siberian wildfires to the Russian Arctic SIC variability is notably intensified during 2004-2021. This does not reveal whether there is a

causal relationship between the two. The cause-and-effect relationship among Siberian blocking events, wildfire events and Arctic warming related to Russian Arctic SIC decline is now more fully discussed in **Methods (Lines 720-757)**.

Calculations further reveal that the correlation coefficient between the FWI index and eastern Siberian Arctic warming varies from 0.33 over 1979-2003 to 0.72 over 2004-2021, even though the eastern Siberian Arctic warming has a negative correlation of -0.12 (-0.74) with the Russian Arctic SIC anomaly over 1979-2003 (2004-2021). Thus, there is a stronger connection of the eastern Siberian Arctic warming to the Russian Arctic SIC change during 2004-2021. Our further analysis reveals that the connection of summer eastern Siberian wildfires to eastern Siberian Arctic warming strongly depends on the extent of Russian Arctic SIC anomaly. When the Russian Arctic SIC decline is weak (strong) as observed during 1979-2003 (2004-2021), the connection of eastern Siberian wildfires to eastern Siberian Arctic warming or SAT anomaly becomes weak (strong). This is a major reason of why we used the Arctic sea-ice instead of the air temperature in our study, even though the air temperature is important for wildfires.

In earlier studies in the literature, Arctic warming (i.e., eastern Siberian Arctic warming) or global warming is often considered as the background condition of wildfire events (e.g., Westerling et al., 2006; Running, 2006; Di Virgilio et al. 2019), even though the accumulation of wildfire event-induced CO₂ emissions can subsequently feedback to Global warming or Arctic warming and leads to ~0.18 °C increase in the global land surface air temperature (Li et al. 2017). Thus, it is highly likely that the Arctic warming as a climatic factor mainly influences high latitude wildfire events probably because the wildfire event-induced global warming trend is weaker compared to the warming trend over eastern Siberia.

Reference: Li, F., et al. Impact of fire on global land surface air temperature and energy budget for the 20th century due to changes within ecosystems. *Environ. Res. Lett.* 12 044014 (2017).

On the other hand, the Arctic warming and sea-ice did not have the same trend over 1979-2003, but have almost the same pattern only over 2004-2021. Because there is a high positive correlation (0.72) between eastern Siberian wildfires and Arctic warming during 2004-2021, we may use the slope or changing rate of the eastern Siberian Arctic SAT time series over 2004-2021 to approximate the trend of the eastern Siberian wildfires over 2004-2021. In our study, we assume that the change rate of the summer-mean eastern Siberian Arctic SAT without Siberian blocking events over 2004-2021 can

be considered as the background Arctic warming (BAW) trend, which is mainly caused by the Russian Arctic SIC decline over 2004-2021. This assumption is well justified because the increase in Arctic warming over eastern Siberia is mainly produced by the Russian Arctic SIC decline, as confirmed by the model experiments of Sato and Nakamura (2019) and Nakamura and Sato (2022).

Sato and Nakamura (2019) analyzed the dataset derived from a 100-member ensemble experiment called the “database for Policy Decision making for Future climate change” (d4PDF) which comprises a 60-year integration (1951–2011) using MRI-AGCM3.2 (Meteorological Research Institute Atmospheric General Circulation Model version 3.2) driven by observed sea surface temperature (SST), sea ice, and natural and anthropogenic forcing. Their results are again shown in Fig. A1 here.

Fig. A1. The first mode governing simulated summer SAT anomalies in d4PDF and its time series.

(a) Regression pattern of JJA-averaged SAT (color), 500 hPa geopotential height (contours: 5m intervals) against the normalized score from each ensemble member (grey line in b) for the first PC. (b) blue and green lines show the JJA means of sea ice area in the Arctic Ocean and North hemispheric SAT, respectively, where here the sea ice area is presented as a percentage change relative the 1951-2010 average (**Taken from Figure 1 of Sato and Nakamura (2019)**).

It is found that the positive summer SAT anomalies over eastern Siberia (Fig. A1a and green lines in Fig. A1b) also increase with the decrease of the sea-ice area (blue line in Fig. A1b), thus demonstrating that the Arctic warming over eastern Siberia can be caused by the decline of the JJA-mean sea-ice because the observed sea-ice is considered as a driving factor in their numerical experiments.

In further support of this perspective, Nakamura and Sato (2022) used six atmosphere-ocean coupled general circulation models to perform Arctic warming experiments for prescribed sea-ice

concentration (SIC) (Fig. A2a) and surface sea temperature (SST) (Fig. A2b) anomalies. The stationary responses of the model ensemble to the prescribed SIC anomaly are also shown in Figs. A2c-d. It is noted that the June-July (JJ) mean 500 hPa geopotential height (GH500) anomaly in response to the prescribed SIC forcing shows a stationary anticyclonic anomaly over eastern Siberia (90°-150°E), which resembles our result (Fig.3f) without Siberian blocking events. The JJ-mean 300-hPa zonal wind anomaly as a response to a prescribed SIC anomaly shows a weakening over eastern Siberia (45°-65°N, 90°-150°E) with positive anomalies in high and lower latitudes, which resembles our result (Fig. 8b) without Siberian blocking events. Thus, in our study, the BAW, stationary anticyclonic anomaly and negative 500-hPa zonal wind anomaly trends over eastern Siberia are mainly produced by the increasing trend of the Russian Arctic SIC.

[REDACTED]

Figure A2. (a) Sea-ice concentration (SIC) and (b) sea surface temperature (SST) anomalies as boundary conditions of perturbation runs (six runs average) from a control run in six atmosphere-ocean coupled general circulation models (CCSM4, CC; GFDL-CM3, GF; HadGEM2-AO, HA; MIROC5, MI; MPI-ESM-MR, MP; and MRI-CGCM3, MR) in the Arctic warming experiment. Differences of perturbation runs (six runs average) minus a control run for June–July mean anomalies of (c) geopotential height (GH) at 500-hPa and (d) zonal wind (U) at 300-hPa (**Taken from Nakamura and Sato (2022)**).

In our revised manuscript (**lines 223-230, 240-244 and 380-384**), we now cite these very relevant works of Sato and Nakamura (2019) and Nakamura and Sato (2022) to support our argument that the eastern Siberian Arctic warming with anticyclonic anomaly and negative zonal wind trend patterns are mainly produced by the increasing decline of the Russian Arctic SIC.

References: Sato, T., and T. Nakamura. Intensification of hot Eurasian summers by climate change and land–atmosphere interactions. *Sci. Rep.*, 9, 10866. (2019).

Nakamura, T., and T., Sato. A possible linkage of Eurasian heat wave and East Asian heavy rainfall in Relation to the Rapid Arctic warming. *Environmental Research*, 209, 112881. (2022).

We also used CESM1 (**see Methods**) to perform our own numerical simulations as discussed used in the revised manuscript.

In response to the Reviewer’s comment, we have analyzed the CESM1 simulations of Dai et al. (2019) and PAMIP ensemble simulations (see the results below and Supplementary Figs. 4-5) to demonstrate that sea-ice loss can enhance summer warming over eastern Siberia. It is noted that the recent rapid loss of the Russian Arctic SIC can strengthen the linkage of the eastern Siberian wildfires to eastern Siberian SAT changes during 2004-2021. In other words, the recent rapid SIC loss can significantly strengthen the warming over eastern Siberia, which in turn increases summer wildfires over eastern Siberia. Although CO₂ and other greenhouse gases (GHGs) have also increased during 2004-2021, which can contribute to the recent warming over eastern Siberia, the magnitude of the GHG increases during this 18-year period is relatively small (~11% for CO₂ increase, see CO₂ time series from NOAA at <https://gml.noaa.gov/ccgg/trends/>) compared with recent rapid sea-ice loss (Fig. 1b).

Note that Supplementary Fig. 4 or Fig. A4 below show that the future warming from increased CO₂ (Supplementary Fig. 4b or Fig. A4b below) over eastern Siberia is much larger than the warming due to SIC loss (Supplementary Fig. 4c or Fig. A4c below). Clearly, their relative contributions to the warming depend on the amount of changes in GHGs and sea ice over the study period. From 2004-2021, GHG increases are relatively small compared with sea-ice loss. This leads us to consider the warming effect of SIC loss as the main contributor to recent warming in the region.

References: Dai, A., D. Luo, M. Song and J. Liu. Arctic amplification is caused by sea-ice loss under increasing CO₂. *Nature Communications*, 10:121 (2019).

On the other hand, because Siberian blocking and wildfire events are of sub-seasonal timescales (≤ 30 days), the slow changes in Siberian blocking and wildfire events may result from the slow modulation of Arctic warming linked to the slow change in Arctic SIC. In this case, we can infer that long-term variations of Siberian blocking and eastern Siberian wildfire events originate from the slow modulation of the Arctic warming linked to the long-term changes in the Russian Arctic SIC over

2004-2021 while the Siberian blocking event produces the wildfire event. The cause-effect relationship between the blocking event or wildfire event and the slow changes in the eastern Siberian Arctic warming associated with the slow variation of the Russian Arctic SIC is discussed in **Methods (lines 720-757)**.

Reference:

Question:

2. Fire data

There is no direct observation of the Siberian region's fire, but it would be better to use remotely sensed fire activity directly instead of FWI because FWI is parameterized based on Canadian fire, not Siberia. As the authors already used GFAS data, I think the main results should be conducted with GFAS data, such as the correlation between GFAS and sea ice. Also, many remote-sensed fire datasets, such as MCD64, GFED, and Fire Radiative Power, are available to compare reality. In addition, a previous study by Rogers et al. 2015, pointed out that fire characteristics in Siberia and North America are distinctly different, so FWI is unsuitable for Siberian fire studies.

Response:

Thank you for these good suggestions which have served to strengthen this part of the paper. Incidentally, we point out that the Fire Weather Index (FWI) data used in our study was taken from the fire danger indices historical data with $0.25^{\circ} \times 0.25^{\circ}$ grid points from the Copernicus Emergency Management Service for the European Forest Fire Information System (EFFIS) (<https://cds.climate.copernicus.eu/cdsapp#!/dataset/10.24381/cds.0e89c522?tab=overview>). The EFFIS incorporates the fire danger indices for three different models developed in Canada, United States and Australia, and is NOT parameterized only using Canadian fire data.

Following your advice, in our revised manuscript we have analyzed the GFAS and MODIS fire data and made a comparison with the FWI results from the EFFIS data. We now show the results from these two sets in Fig. 2. One can see that the wildfire trends over eastern Siberia over 2004-2021 based on the GFAS and MODIS data (Fig. 2) are consistent with the FWI trend (Fig. 1). A correlation analysis shows that during 2004-2021 the FWI has a negative correlation of -0.56 with the Russian Arctic SIC, whereas the wildfire fraction over eastern Siberia has a negative correlation of -0.55 with the Russian Arctic SIC over 2004-2021. This establishes that the FWI data is suitable for studying wildfires over eastern Siberia. We also note that for Siberian blocking events the composite daily results based on

the daily wildfire fraction of the GFAS data (Fig. 6) are basically consistent with the composite daily FWI results from the EFFIS data (Fig. 5).

Figure A3. (a) Normalized time series of summer (June to August, JJA) mean burned area anomaly averaged over eastern Siberia (90°-150°E; 60°-75°N) during 1997-2016 based on the Global Fire Weather Database (GFWED). **(b)** Linear trend patterns of JJA-mean burned fraction (nondimensional) anomaly over 1997-2016 for the GFWED, where the dots denote regions over which the trends are significant at the 95% confidence level in a two-sided student’s t-test.

We further show the normalized time series of summer mean burned area anomaly averaged over eastern Siberia (90°-150°E; 60°-75°N) during 1997-2016 and its linear trend pattern over 2004-2016 in Fig. A3 based on the Global Fire Weather Database (GFWED). It is found that summer eastern Siberian wildfires show a notable increasing trend over 2004-2016, in agreement with the results from the EFFIS, GFAS and MODIS data. Thus, the result about the increasing trend of summer wildfires over eastern Siberia over 2004-2021 for the FWI of the EFFIS data used in our study can be supported

by the GFAS, MODIS and GFWED data. Also, some comparisons of the FWI data with the GFAS and MODIS data are added in **lines 112-141** in our revised manuscript.

In the discussion section of our revised manuscript, we have cited the paper of Rogers et al. (2015) and presented some of the caveats to our investigation. Thank you for alerting us to this paper. It presents the important finding that tree species have a significant impact on the wildfires over Siberia and North America. We are familiar with the findings of the more recent paper on this topic by Webb, E.E., Alexander, H.D., Paulson, A.K., Loranty, M.M., DeMarco, J., Talucci, A.C., Spektor, V., Zimov, N. and Lichstein, J.W., 2024: Fire-induced carbon loss and tree mortality in Siberian larch forests. *Geophys. Res. Lett.*, **51**, e2023GL105216, doi: 10.1029/2023GL105216 and have also referenced it in this context.

Question:

3. Global climate (Earth system) model simulation is needed.

If authors want to provide solid evidence of sea ice's role in fire and blocking, I think authors need to run climate model (Earth system model) simulations to provide differences between sea ice decline simulations and control simulations. Then, it clearly finds how sea ice loss occurs by blocking and fire-prone weather, but only correlation analysis cannot provide enough background.

Response:

We appreciate this suggestion of supporting our results with appropriate climate model simulations. These integrations have truly served to reinforce our message. In the revised manuscript, the Arctic warming over eastern Siberia is considered as the sum of the background Arctic warming associated with the slowly varying Russian Arctic SIC decline and slow changes in Siberian blocking events, even though the slow variation of Siberian blocking comes from the slow modulation of the background fields based on the blocking theory (Luo et al. 2019, Zhang and Luo 2020). Such an assumption justifies because the Siberian blocking or wildfire events are often of sub-seasonal timescales (≤ 30 days). To some extent, the slow change in the Arctic warming associated with the sea-ice loss can be considered as a background condition influencing Siberian blocking or wildfire events. The blocking theory we used here has been described in **Methods (Lines 792-808)**.

In our revised manuscript, we analyzed the CESM1 simulations of Dai et al. (2019) to provide differences between simulations with and without sea-ice loss. We showed the JJA-mean SAT changes for the 1% CO₂ runs with (1%CO₂) and without (FixedIce) fixed sea-ice and their difference averaged

over the 50 years around the time of the second CO₂ doubling from 116-165 years in Fig. A4 below.

Figure A4. (a-c) CESM1-simulated JJA-mean SAT changes (°C, relative to pre-industrial control run) averaged over 45°-90°N and over the 50 years around the time of the second CO₂ doubling from 116-165 years for **(a)** the 1% CO₂ run with fully interactive sea ice (denoted as 1% CO₂ run) and **(b)** the 1% CO₂ run with fixed SIC in surface flux calculations (denoted as FixedIce run) and **(c)** their difference, Shaded areas indicate a statistical significance at the 5% level based on a two-sided students t-test.

It is noted that the difference between two experiments shows a positive anomaly in high latitudes of Eurasia (**Fig. A4c**). Thus, the SIC loss can lead to enhanced Arctic warming over eastern Siberia. Consequently, our CESM1 simulations suggest that the recent rapid loss of Russian Arctic SIC (**Fig. 1b**) can cause stronger Arctic warming over eastern Siberia than the case without SIC loss. Given the relatively small GHG increases during 2004-2021 (see discussion above), we think the SIC-induced warming is likely to dominate during this recent period. The CESM1 simulation results are given in **Supplementary Fig. 4**.

For a comparison, we also analyzed the PAMIP ensemble simulations with specified SIC changes based on CMIP6 models, focusing the JJA-mean SAT difference between the atmospheric model simulations with future and pre-industrial SIC but with the same SST (**Fig. A5**).

Fig. A5. JJA-mean SAT difference (°C) between the PAMIP atmospheric model simulations with specified future and pre-industrial Arctic sea-ice concentrations (SIC) from the Polar Amplification Model Intercomparison Project (PAMIP) ensemble of CMIP6 models (Smith et al. 2019). The PAMIP ensemble consists of 1000 runs from 10 models. The color shading represents the region with the 5% confidence level based on a two-sided students t-test.

It is found that the PAMIP ensemble result based on CMIP6 models (**Fig. A5 or Supplementary Fig. 5**) is consistent with the CESM1 simulation result (**Fig. A4**) with enhanced summer warming over eastern Siberia due to the SIC loss. Thus, the recent rapid loss of the Russian Arctic SIC can lead to enhanced Arctic warming over eastern Siberia even though sub-seasonal atmospheric circulation patterns (e.g., Siberian blocking events) are absent. This simulation result is consistent with those of Sato and Nakamura (2019) and Nakamura and Sato (2022). Thus, the background Arctic warming over eastern Siberia due to the SIC loss can be considered as the climatic background condition of Siberian blocking or eastern Siberian wildfire events. In fact, because the climate models cannot realistically simulate the blocking frequency (Davini and D’Andrea, 2020), it is difficult to use the climate models to identify the contributions from the Russian Arctic SIC decline and internal atmospheric variability

(i.e., changes in Siberian blocking events) to the increasing trend of eastern Siberian wildfires over 2004-2021. In our manuscript (**Lines 186-191 and Methods**), we present a new method to approximately estimate the different contributions of the Arctic warming and changes in Siberian blocking events as described in **Methods (lines 758-791)**.

Response to Reviewer #2's comments for the manuscript (NCOMMS-23- 54142)

We very much appreciate the helpful comments of Reviewer #2. We have substantially revised our paper taking cognizance of your comments (in italics). Our point-by-point responses are presented below (regular font):

Response to general statement:

Based on the meteorological and satellite data, this study has revealed most of the surge in eastern Siberian wildfires from 2004 to 2021 is attributable to the background Arctic warming linked to the decline in Russian Arctic sea ice. The remaining is associated with internal atmospheric variability linked to changes in Siberian blocking events. Furthermore, over the period 2004-2021, Siberian blocking events at higher latitudes exhibit increased persistence and larger zonal scales compared to the period 1979-2003, attributed to reduced meridional potential vorticity gradients resulting from background Arctic warming. This topic is interesting and highly original. The author has provided a new perspective to understand the recent increases in eastern Siberian wildfires. I thought that this manuscript has potential to be published in Nature Communication. Thus, I recommend that the manuscript needs minor revision.

Response:

We appreciate your very encouraging comments for improving the manuscript. We have made a minor revision based on your useful comments.

Response to Minor comments:

I, This article, in my view, places more emphasis on revealing the connection between sea ice, blocking, and wildfires with high temperatures and circulation patterns. In fact, high temperature is one of the most important elements. However, based on the FWI, besides of temperature, the relative humidity, precipitation and wind can be also contributed to the variations of FWI. It lacks a thorough demonstration of the specific processes leading to wildfires, relying heavily on literature references. I suggest that authors can add some discussions about the impacts and contributions of these potential factors.

Response:

Thanks for your constructive comments. In this revised manuscript, we have added further relevant considerations in the Discussion section as to the impacts and contributions of many potential factors that influence wildfires. While the high air temperature or land surface warming and decreased precipitation are two key factors influencing wildfires, other factors such as soil moisture extent, snowpack, vegetation coverage, frozen soil, tree species, lightning, hydrological cycle and fuel flammability can influence the variability and trend of wildfires. However, the soil moisture, vegetation coverage, lightning and fuel flammability can also be altered by land surface warming and precipitation changes via atmosphere-land-ecosystem coupling. Thus, it is reasonable to calculate the impact of eastern Siberian Arctic warming and precipitation on the Siberian wildfires, even though the wildfire is a result of the combined effect of multiple weather and climate factors as well as multi-scale interaction processes. On **lines 435-458** of our revised manuscript, we have added some discussions about the possible impacts of the other factors.

2, Authors have mentioned AMO, PDO, blockings, with different time scales. As investigating the causes of wildfires, particularly as extreme events, special attention should be given to the issue of matching time scales.

Response:

In the revised manuscript, we have emphasised that AMO and PDO have decadal timescales (\geq 10 years), while the Siberian blocking has a timescale of 10-20 days. However, it is the long-term or decadal changes in Siberian blocking that is relevant for the trend of wildfire changes. Thus the AMO and PDO can influence Siberian blocking events on decadal time scales via altering the background condition because of the slow variation of eastern Siberian Arctic warming related to Russian Arctic SIC variability modulated jointly by the AMO and PDO. Because Siberian blocking events can lead

to a Siberian wildfire events through generating heatwaves over eastern Siberia, the long-term variation of Siberian wildfires can have a footprint of AMO and PDO through their impact on blocking. This may be a major cause of why the eastern Siberian SAT time series with and without Siberian blocking events have different change rates. In **lines 435-439 and in methods (lines 720-757)**, we have added some text to discuss the timescale issue about the slow variability of the Russian Arctic SIC and Siberian blocking, even though the slow variation of the Russian Arctic SIC is linked to AMO and PDO.

3, Line 92, authors mentioned using a novel approach. Please clarify the “new” in the approach.

Response:

In the Methods part (**lines 758-791**) of our revised manuscript, we have described our novel approach to clarify why our approach is new.

4, line 215 How to understand 84%, a very high percentage?

Response:

Yes. It indicates that the background Arctic warming plays a major role in determining the increasing trend of wildfires.

REVIEWER COMMENTS

Reviewer #1 (Remarks to the Author):

Luo et al. revised the manuscript based on several major concerns from reviewers, but I still have a few major points that should be addressed by the authors.

1. FWI sensitivity to seasonal mean temperature and/or others.

Authors argue that summer background Arctic warming (BAW) over eastern Siberia can enhance fire risk. However, FWI is not based on seasonal mean temperature; it is actually based on daily maximum temperature. Arctic amplification definitely, on the seasonal time scale, can contribute to increased maximum temperature as well, but the manuscript does not contain any related information.

Even if there is a clear relationship between BAW and maximum temperature, it is still unclear how sea ice modulates FWI in case blocking happens. As FWI is based on four meteorological conditions: maximum temperature, relative humidity, precipitation, and wind speed, authors need to quantify contributions from each component or their relevance by comparing them. For example, temperature increases will lead to higher saturated evaporation, which reduces relative humidity, given the amount of specific humidity. In contrast, precipitation is reduced during blocking events, while convective activity will be increased by higher surface temperature and enhanced instability in the lower atmosphere.

2. Blocking mechanism

The authors did a composite analysis comparing blocking events under high and low sea ice conditions. This analysis gives many insights but Figures 5 to 7 have similar messages. (Also, Figures 1 to 4 have similar patterns, so I was wondering if authors could reduce the number of figures and show them compactly.)

Furthermore, authors may analyze raw anomaly time series instead of de-trend datasets. Then, high SIC cases have been picked up in recent years, so it would not easily remove other effects besides SIC conditions. Authors need to provide analyzing results with all de-trended datasets, not only SIC but also all other variables, even for separating blocking events.

Also, the authors did not provide details on how many cases were picked up for composite analysis. Finally, as the authors provided CMIP6-PAMIP and CESM1 simulations, it is need to do same composite analysis (-10 lag day to 10 lead day) for all variables, including FWI (can be calculated with four meteorological variables), SAT, Z500, precipitation, and PVy. Then, the authors may clearly explain how BAM and SIC modulate stronger blocking systems to enhance the risk of fire in Siberia.

3. Fire characteristic

In Figure 2, it is clearly seen that wildfire fraction (GFAS?) and burned trend are totally different. It might be that GFAS does not contain any information related to plant function type (taiga forest, needleleaf/broadleaf, deciduous/evergreen, moss/lichen, and barren), while MODIS is regarded as direct satellite observation. In other words, the authors need to explain why only regions in the red

color of Figure 2d have a clear linear trend in the burned area.

The authors justified that there were no available datasets before the MODIS-era. Still, authors need to use longer data, such as AVHRR-based long-term datasets, especially to discuss the decadal change in fire risk.

<https://catalogue.ceda.ac.uk/uuid/62866635ab074e07b93f17fbf87a2c1a>

For interannual variability, I believe 23 years under MODIS coverage is enough to understand the relationship with climate conditions.

Minor comments:

1. Units are missing in all figure's captions. The unit for trend should be present per year or decade.
2. Use the minus symbol instead of a short dash.
3. The authors used color bars with continuous coloring, but it is not easy to check the exact value in the figure. May change to 11 or 15 levels in color bars.

Response to Reviewer #1's comments for the manuscript (NCOMMS-23- 54142)

We appreciate the constructive comments of Reviewer #1. We have made substantial revisions according to his/her remarks. These are presented below in italic text and our responses are given in plain text:

General question:

Luo et al. revised the manuscript based on several major concerns from reviewers, but I still have a few major points that should be addressed by the authors.

Response:

Thank you for your useful suggestions. We have further made a major revision according to your suggestions. In particular, the remarks you made in connection with the four meteorological parameters involved with the FWI have led us to think more deeply about this complex issue. Recent research presented in the literature (e.g., Seager et al., 2015; Zhuang et al., 2021; Clarke et al. 2022) has shown that vapor pressure deficit (VPD) is ‘the leading meteorological variable that controls wildfires’, and this new perspective is perfectly consistent with our results. See below for a more detailed argument on this.

Response to Major comments:

1. FWI sensitivity to seasonal mean temperature and/or other factors.

Question:

Authors argue that summer background Arctic warming (BAW) over eastern Siberia can enhance fire risk. However, FWI is not based on seasonal mean temperature; it is actually based on daily maximum temperature. Arctic amplification definitely, on the seasonal time scale, can contribute to increased maximum temperature as well, but the manuscript does not contain any related information.

Response:

In most analyses Arctic warming is calculated from daily-mean SAT, whereas FWI is calculated in terms of the input of the daily noon (or maximum) air temperature (T_{\max}), relative humidity, wind speed, precipitation, drought conditions, fuel availability, vegetation characteristics and topography (Fire Weather Index - Monthly Mean, 1979-2020 —Discover the key services, thematic features and tools of Climate-ADAPT (europa.eu)). By calculating the JJA-mean value of daily FWI, one can obtain the seasonal mean FWI. Another perspective on this is that the FWI calculated with a seasonal mean

temperature can give an indication of fire risk in that season.

In the revised manuscript, we have compared the difference of the result between daily maximum and daily-mean SATs. When the daily maximum SAT is used, the obtained JJA-mean Arctic warming over eastern Siberia has a high correlation of 0.99 ($p < 0.01$) with the JJA-mean Arctic warming from daily-mean SAT over 1979-2021. They also have the same correlation coefficient of 0.99 over 1979-2003 and 2004-2021. Thus, using both the daily maximum and daily-mean SAT data does not seriously distort the summer Arctic warming. We also discuss whether using daily-mean and daily maximum SATs influence the correlation between the summer Arctic warming and FWI. We find that the summer (JJA-mean) FWI and SAT over eastern Siberia have a positive correlation coefficient of 0.72 ($p < 0.01$) (0.33, $p < 0.05$) over 2004-2021 (1979-2003) for non-detrended data if the daily-mean SAT is used. However, their correlation becomes 0.72 ($p < 0.01$) (0.34, $p < 0.05$) over 2004-2021 (1979-2003) if the daily maximum SAT is used. Clearly, the correlation between summer FWI and Arctic warming is not altered by using daily maximum SAT or using daily-mean SAT. We have briefly described this point on **pages 232-239**.

Question:

Even if there is a clear relationship between BAW and maximum temperature, it is still unclear how sea ice modulates FWI in case blocking happens. As FWI is based on four meteorological conditions: maximum temperature, relative humidity, precipitation, and wind speed, authors need to quantify contributions from each component or their relevance by comparing them. For example, temperature increases will lead to higher saturated evaporation, which reduces relative humidity, given the amount of specific humidity. In contrast, precipitation is reduced during blocking events, while convective activity will be increased by higher surface temperature and enhanced instability in the lower atmosphere.

Response:

In our revision we now clearly describe how sea-ice loss modulates wildfires via changes in Siberian blocking. Our physical chain of reasoning is that summer sea-ice loss leads to Arctic warming via radiative heating (Pistone et al. 2019), and Arctic warming as a background tends to reduce PV_y . This, in turn, results in increased persistence, larger zonal scale and reduced eastward movement of Siberian blocking as described in Fig. 8a as a schematic diagram and on **lines 440-462**, and hence enhanced heatwaves and wildfire events via reducing energy dispersion and enhanced nonlinearity of

blocking system. If PV_y exhibits a decrease, Siberian blocking and the associated vapor pressure deficit (VPD) will show increases in lifetime and zonal scale as well as a decrease in its eastward movement speed, thus encouraging wildfires. Therefore, changes in Siberian blocking events combine to amplify wildfires over eastern Siberia through increased VPD and reduced precipitation.

To elaborate, while FWI is based on the four meteorological conditions of maximum temperature, relative humidity, precipitation, and wind speed, it is difficult (and may not even be meaningful to try) to separate the individual contributions to changes in the summer FWI from the four meteorological parameters because these four parameters are coupled together and not independent of each other. For example, the increase in air temperature due to Arctic sea-ice loss happens in Siberia via radiative heating due to reduced top-of-atmosphere albedo as the sea-ice is melting (Pistone et al. 2019), which increases the saturation vapor pressure and hence reduces the relative humidity. At the same time, the increased air temperature leads to reduced precipitation (Fig. 3c) and changes in wind speeds (Fig. 7b, supplementary Fig. 9). We stress that in the manuscript, our emphasis is to evaluate the relative contributions of the BAW and Siberian blocking events, rather than four meteorological variables, to increases in eastern Siberian wildfires.

As pointed out in many previous studies (e.g., Seager et al. 2015; Zhuang et al. 2021; Blach et al. 2022), the VPD, which includes air temperature and relative humidity (or saturation vapor pressure), is a key meteorological factor that controls wildfires. Because VPD can explain more of the variance in fire activity than can precipitation, drought indices, air temperature, or wind individually, it is more successful in explaining the ignition, spread, intensity and size of forest fires than other meteorological variables. In the revised manuscript, we use VPD instead of other meteorological variables. Then we use daily-mean VPD to examine the contributions from the BAW and changes in Siberian blocking events to increases in eastern Siberian wildfires by calculating the slope rates of the JJA-mean VPD with and without Siberian blocking events (Fig. 4 for daily-mean SAT and Supplementary Figure 8 for daily maximum SAT).

Our method based on calculating the slope rates (i.e., trends) of the JJA-mean VPD with and without Siberian blocking events is a new approach, which has not used in previous studies. Moreover, in our revised submission we also discussed the difference of the result between daily maximum and daily-mean SATs. For example, the JJA-mean VPD from the daily-mean SAT has a positive correlation of 0.98 with that from the daily maximum SAT. In other words, the VPD-SAT relationship is not

significantly influenced by using the daily maximum or daily-mean SAT, even though the linear trends of VPD based on daily-mean and maximum SATs are slightly different. This has been discussed in **lines 275-298 and 316-338**.

In our revised manuscript, we see that the increasing trend of summer VPD is dominated by the BAW because of reduced precipitation and enhanced saturated vapor pressure. When the lifetime, zonal scale and reduced eastward movement of Siberian blocking is favored by the BAW via reducing the meridional potential vorticity gradient (PV_y), more persistent increases (decreases) in air temperature (precipitation) can be seen during the blocking episode (Figs. 5-6) due to longer-lasting subsidence and short-wave radiation heating via the blocking maintenance, which lead to more persistent increases in VPD and wildfire events. During the blocking episode with a large-scale ($\sim 5000\text{km}$ or larger), the increase in air temperature can occur due to persistent subsidence, short-wave radiation heating and temperature advection from lower latitudes throughout the troposphere because of the long persistence of summer blocking high (Drouard and Woollings 2018). This is a well-known blocking effect. However, the higher surface air temperature has different effect for a convective or synoptic-scale ($\leq 1000\text{km}$) system. In a small-scale system, convective activity and convective instability can take place. As a result, in this system convective activity can be increased by higher surface temperature and enhanced instability in the lower atmosphere. However, this convective activity or instability does not easily take place in a large-scale blocking anticyclone system, even though convective available potential energy is intensified (Chen et al. 2021).

1. Seager, R. et al. Climatology, variability, and trends in the us vapor pressure deficit, an important fire-related meteorological quantity. *J. Appl. Meteorol. Climatol.* 54, 1121–1141. (2015)
2. Zhuang et al. Quantifying contributions of natural variability and anthropogenic forcings on increased fire weather risk over the western United States, *Proceedings of the National Academy of Sciences of the United States of America*. 118 (45) e2111875118 (2021).
3. Blach, J. et al. Warming weakens the night-time barrier to global fire, *Nature*, 602, 442-448. (2022)

4. Drouard, M., & Woollings, T. (2018). Contrasting mechanisms of summer blocking over western Eurasia. *Geophysical Research Letters*, 45, 12,040–12,048.
5. Chen, Y., Romps, D. M., Seeley, J. T., Veraverbeke, S., Riley, W. J., Mekonnen, Z. A., & J. T. Randerson, Future increases in Arctic lightning and fire risk for permafrost carbon. *Nature Climate Change*, 11, 404–410. (2011).

2. Blocking mechanism

Question:

The authors did a composite analysis comparing blocking events under high and low sea ice conditions. This analysis gives many insights but Figures 5 to 7 have similar messages. (Also, Figures 1 to 4 have similar patterns, so I was wondering if authors could reduce the number of figures and show them compactly.)

Response:

We appreciate the spirit of this comment, and we have re-thought the presentation of the Figures. Now Fig. 1 establishes a linkage of eastern Siberian wildfires to Russian Arctic SIC using FWI and sea-ice, whereas Arctic warming over eastern Siberia and its trends over two sub-periods are described in Fig.2. Fig. 3 describes the variability of precipitation and its linear trends over 1979-2003 and 2004-2021. Fig. 4 presents the variability of vapor pressure deficit (VPD) and its linear trends over 1979-2003 and 2004-2021. Thus, it is reasonable to show these results in Figs. 1-4 because they reflect different meteorological variables. Then, by comparing the difference of VPD with and without Siberian blocking events, one can estimate the different contributions of the BAW and changes in

Siberian blocking events to the increase in eastern Siberian wildfires using the VPD based on daily-mean and daily maximum SATs.

In Figs.5-6, our attention is focused on examining how the daily SAT, precipitation and VPD depend on the evolution of Siberian blocking events from a daily composite. They can tell us what are the significant changes in daily SAT, precipitation and VPD for wildfire events during the blocking evolution.

Question:

Furthermore, authors may analyze raw anomaly time series instead of de-trend datasets. Then, high SIC cases have been picked up in recent years, so it would not easily remove other effects besides SIC conditions. Authors need to provide analyzing results with all de-trended datasets, not only SIC but also all other variables, even for separating blocking events.

Response:

Thank you for suggesting the potential insights that might be gained from considering the raw anomaly time series. In the interests of keeping the paper within a reasonable size we have decided not to include these extra analyses. We believe the detrended data provide valuable information as to the interannual variability.

As to the point you raise in the last sentence above, we apologize for our lack of clarity in describing the data preparation. In fact, the SIC and all other variables were detrended before performing our daily composite of Siberian blocking events. This is now explicitly stated in the revised manuscript (**lines 381-384**).

Question:

Also, the authors did not provide details on how many cases were picked up for composite analysis.

Response:

Thanks, we now provide this information (**lines 390-392**).

Question:

Finally, as the authors provided CMIP6-PAMIP and CESM1 simulations, it is need to do same composite analysis (-10 lag day to 10 lead day) for all variables, including FWI (can be calculated with four meteorological variables), SAT, Z500, precipitation, and PVy. Then, the authors may clearly explain how BAW and SIC modulate stronger blocking systems to enhance the risk of fire in Siberia.

Response:

Thanks for this suggestion for added analysis of the CMIP6-PAMIP and CESM1 simulations. However, we feel that the tailored results we have presented in connection with these runs establish the veracity of the case we are making. We don't think that performing all these lead-lag composites for the five variables is required or would really add to the understanding. Making the paper considerably longer in this respect would not be warranted.

However, in response to the spirit of your comment, in our revision we have some discussion (**lines 440-462**) and added a schematic diagram (Fig. 8a) to explain how BAW and SIC modulate stronger blocking systems to enhance the risk of fire in Siberia. Moreover, we further use two sets of experiments with 100 members of AWI-CM-1-1-MR in CMIP6-PAMIP to analyze the impact of Arctic warming or Arctic amplification (AA) on summer Siberian blocking events. One is forced by present-day sea surface temperature (SST) and present-day SIC (pdSST-pdArcSIC, hereafter PD) and the other is forced by present-day SST and future SIC (pdSST-futArcSIC, hereafter FU). The SST is the 1979–2008 climatology from the Hadley Center Sea Ice and SST dataset, as are the SIC condition

in PD. In FU, the SIC condition is set as the future SIC under 2°C average global warming background from CMIP5 RCP8.5. Each experiment is run for 14 months with the first 2 months are ignored as spin-up, and each model produces 100 ensemble members per run. More detail can be found in Smith et al. (2019).

Using the TM index of blocking events, we find that there are 131 (116) Siberian blocking events in the region (90°-120°E) in summer for PD (FU). We perform daily composites of Z500, SAT and precipitation of 131 (116) Siberian blocking events for PD (FU) and show their results in Fig. A2.

Figure A1. Time-longitude evolutions of composite daily (a, b, c) 500-hPa geopotential height (Z500), (d, e, f) surface air temperature (SAT) and (g, h, i) precipitation anomalies averaged over 60°-70°N of 131 (116) Siberian blocking events for (a, d, g) PD and (b, e, h) FU as well as (c, f, i) their differences.

It is found that Siberian blocking is longer-lived and has slightly larger zonal scale and lesser eastward movement for FU (Fig. A1b) than for PD (Fig. A1a). Their Z500 difference (FU minus PD) is shown in Fig. A1c. Similarly, the daily warming as denoted by the positive SAT anomaly is more persistent for FU (Fig. A1e) than for PD (Fig. A1d). The FU minus PD SAT difference (Fig. A1f) clearly shows that the variation of the daily SAT from PD to FU (Fig. A1f) during the blocking episode exhibits a high similarity with that of the daily Z500 (Fig. A1c). We also see that during the blocking process the reduction of daily precipitation is more persistent for FU (Fig. A1h) than for PD (Fig. A1g). The variation of the FU minus PD precipitation difference (Fig. A1i) has a high consistency with that of the SAT (Fig. A1f) or Z500 (Fig. A1c). Thus, the Siberian blocking is more persistent and has larger zonal scale and lesser eastward movement under FU than under PD. This shows that Arctic sea-ice loss is conducive to enhanced warming and reduced precipitation during a blocking episode. This model result supports our finding from ERA5, even though climate models do not simulate well blocking events and associated extreme precipitation events.

We also calculate the difference of SAT and PVy without Siberian blocking events between FU and PD from the AWI-CM-1-1-MR simulation to see whether the summer warming and PVy create a favorable condition for Siberian blocking. We show their results in Fig. A2.

Figure A2. Difference of JJA-mean SAT and meridional potential vorticity gradient (PVy) between FU and PD based on AWI-CM-1-1-MR. The dots represent the area with 90% confidence level.

We can see that summer Arctic warming occurs in a more widespread region from Barents Kara Sea (BKS) to eastern Siberia for FU than for PD (Fig. A2a). As a result, JJA-mean PVy is reduced on the south side of Arctic warming for FU than for PD (Fig. A2b). Thus, Arctic sea-ice loss can reduce JJA-mean PVy to influence Siberian blocking events through Arctic warming as shown in Fig. 8a. Consequently, the Arctic sea-ice loss favors increased persistence, zonal scale, reduced eastward movement and increased slow decay of atmospheric blocking due to reduced PVy via reducing energy dispersion and strengthening nonlinearity. In brief, Siberian blocking favors more intense warming and lesser precipitation that increase the likelihood of wildfire events during the blocking episode under a lower sea-ice condition. The model simulation results support our findings from ERA5. However, as

commented above, we do not place the composite results in our manuscript because of the page length limit. Instead, we only present the data result in our manuscript.

References:

Martinez-Villalobos, C. and J. D. Neelin. Climate models capture key features of extreme precipitation probabilities across regions, *Environ. Res. Lett.* 16, 024017 (2021).

Davini, P. and F. D'Andrea. From CMIP3 to CMIP6: Northern Hemisphere Atmospheric Blocking Simulation in Present and Future Climate. *J. Climate*, 33, 10021-10038 (2020).

Question:

3. Fire characteristic

Question:

In Figure 2, it is clearly seen that wildfire fraction (GFAS?) and burned trend are totally different. It might be that GFAS does not contain any information related to plant function type (taiga forest, needleleaf/broadleaf, deciduous/evergreen, moss/lichen, and barren), while MODIS is regarded as direct satellite observation. In other words, the authors need to explain why only regions in the red color of Figure 2d have a clear linear trend in the burned area.

Response:

The Global Fire Assimilation System (GFAS) assimilates fire radiative power observations from satellite-based sensors to produce daily estimates of biomass burning emissions. It includes: Fire Radiative Power (FRP), dry matter burnt and biomass burning emissions. FRP observations currently assimilated in GFAS are the NASA Terra MODIS and Aqua MODIS active fire products (<http://modis-fire.umd.edu/pages/ActiveFire.php>). It has been extended to include information about injection heights derived from fire observations and meteorological information from the operational weather

forecasts of ECMWF. The global Fire Emission Database (GFED) has a 0.25-degree resolution, which combines satellite information on fire activity and vegetation productivity to estimate gridded monthly burned area and fire emissions as well as scalars that can be used to calculate higher temporal resolution emissions.

Here we show the results from Global Fire Assimilation System (GFAS), MODIS and GFED to confirm their consistency regarding the increasing trend of wildfires occurring over eastern Siberia over 2004-2021 (Fig. A4), even though the GFAS, MODIS and GFED data include slightly different information. In Fig. A4, we see that the regions of increased wildfire fraction and burned area over 2004-2021 are consistent between GFAS and MODIS, even though the definitions of wildfire fraction and burned area are different. The areas of increased eastern Siberian wildfires are not totally different between GFAS and MODIS. These data results have been moved to the Supplementary Figures file.

As seen from Fig. A4, the increasing trend of burned fraction over 2004-2016 from GFED mainly appears in the eastern Siberia (Fig. A4f), also consistent with the burned area trend over 2004-2020 from MODIS (Fig. A4d) and wildfire fraction trend over 2004-2021 from GFAS (Fig. A4b), indicating that wildfires over eastern Siberia indeed show a significant increase after 2004, in agreement with the FWI trend (Fig. 1e). Consequently, the result about the increasing trend of eastern Siberian wildfires is not sensitive to the data choice. In Fig. A4d, the red color mainly represents an increase and thus has a clear linear trend in the burned area because the increase in surface warming and precipitation deficit are most notable in eastern Siberia over 2004-2021 (Fig.2e and Fig.3c). Our purpose in the paper is not to examine the cause of the difference among different GFAS, MODIS and GFED, but rather to evaluate the relative contributions of the BAW and changes in Siberian blocking events to increases in eastern Siberian wildfires.

Figure A4. (a, c, e) Normalized time series of summer (June to August, JJA) mean wildfire fraction and burned area anomalies averaged over eastern Siberia (90°-150°E; 60°-75°N) during 2003-2021 based on the (a) GFAS, (c) MODIS and (e) GFED data. (b, d, f) Linear trend patterns of JJA-mean (b) wildfire fraction (unit: non-dimensional value per decade) over 2004-2021, (d) burned area (unit: m^2 per decade) over 2004-2020 and (f) burned fraction (unit: m^2 per decade) over 2004-2016 for the (b) GFAS, (d) MODIS and (f) GFED data, where the dot represents the significant region above the 95% confidence level for a two-sided student's t-test.

Question:

The authors justified that there were no available datasets before the MODIS-era. Still, authors need to use longer data, such as AVHRR-based long-term datasets, especially to discuss the decadal change in fire risk. <https://catalogue.ceda.ac.uk/uuid/62866635ab074e07b93f17fbf87a2c1a>

Response:

Thank you for the suggestion. The AVHRR - LTDR Grid v1.1 product dataset provides monthly information on global burned area on a 0.25 x 0.25 degree resolution grid from 1982 to 2018, although we note that the year 1994 is omitted as there was not enough input data for this year. The dataset includes 4 layers: sum of burned area, standard error, fraction of burnable area and fraction of observed area.

Here we show the linear trend of the JJA-mean burned area over 2004-2018 from the AVHRR – LTDR data in Fig. A4. It is seen that a significant increasing trend of burned area over 2004-2018 mainly appears in eastern Siberia, consistent with those from the FWI, GFAS, MODIS and GFED. Although these data have different time lengths, we stress that in our manuscript we mainly emphasize the linear trend of wildfires over eastern Siberia over 2004-2021 and examine different contributions of the BAW and changes in Siberian blocking events to increases in eastern Siberian wildfires over 2004-2021. As discussed above, many previous studies have established that VPD is a very insightful parameter for revealing the contributions of natural variability and anthropogenic forcings to wildfires (Zhuang et al., 2021; Blach et al. 2022). Thus, in our revised manuscript we use VPD instead of FWI, GFAS, MODIS, GFED and AVHRR – LTDR data to examine our problem. By calculating daily VPD with and without Siberian blocking events, we can estimate different contributions of the BAW and changes in Siberian blocking events to increases in eastern Siberian wildfires over 2004-2021. This approach is one of the novel features of our paper.

Figure A5. (a) Normalized time series of summer (June to August, JJA) mean burned area averaged over eastern Siberia (90°-150°E; 60°-75°N) during 1995-2018 based on the AVHRR – LTDR data. (b) Linear trend patterns of JJA-mean (b) burned area wildfire fraction (unit: 10^6 m^2 per decade) over 2004-2018, where the dot represents the significant region above the 95% confidence level for a two-sided student's t-test.

Question:

For interannual variability, I believe 23 years under MODIS coverage is enough to understand the relationship with climate conditions.

Response:

In the manuscript we make it clear that we did not consider the role of interannual variability because our purpose in this study is focused on examining the linear trend of wildfires over the 2004-2021 period.

Response to minor comments:

Question:

1. Units are missing in all figure's captions. The unit for trend should be present per year or decade.

Response:

In this revised manuscript, we have added units in all Figure captions.

Question:

2. Use the minus symbol instead of a short dash.

Response:

Thanks. In all figure captions and texts, we have now used minus symbol instead of a short dash

Question:

3. The authors used color bars with continuous coloring, but it is not easy to check the exact value in the figure. May change to 11 or 15 levels in color bars.

Response:

Thanks for the comment. In the revised manuscript, we have changed color bars with continuous coloring in Figs. 5-6 into color bars with 11 levels.

REVIEWERS' COMMENTS

Reviewer #1 (Remarks to the Author):

Responses from authors are enough to solve my questions and concerns, so I do not have any further matters to assign for accepting this manuscript in Nature Communications.